# Achieving single nucleotide sensitivity in direct hybridization genome imaging

Yanbo Wang[1,10], W. Taylor Cottle[1,10], Haobo Wang[2], Momcilo Gavrilov [1], Roger S. Zou[3], Minh-Tam Pham [4,5,6,7], Srinivasan Yegnasubramanian [5,6,7], Scott Bailey[1,2] & Taekjip Ha [1,3,8,9] ✉

Direct visualization of point mutations in situ can be informative for studying genetic diseases and nuclear biology. We describe a direct hybridization genome imaging method with single-nucleotide sensitivity, single guide genome oligopaint via local denaturation fluorescence in situ hybridization (sgGOLDFISH), which leverages the high cleavage specificity of eSpCas9(1.1) variant combined with a rationally designed guide RNA to load a superhelicase and reveal probe binding sites through local denaturation. The guide RNA carries an intentionally introduced mismatch so that while wild-type target DNA sequence can be efficiently cleaved, a mutant sequence with an additional mismatch (e.g., caused by a point mutation) cannot be cleaved. Because sgGOLDFISH relies on genomic DNA being cleaved by Cas9 to reveal probe binding sites, the probes will only label the wild-type sequence but not the mutant sequence. Therefore, sgGOLDFISH has the sensitivity to differentiate the wild-type and mutant sequences differing by only a single base pair. Using sgGOLDFISH, we identify base-editor-modified and unmodified progeroid fibroblasts from a heterogeneous population, validate the identification through progerin immunofluorescence, and demonstrate accurate sub-nuclear localization of point mutations.

Single-nucleotide variation (SNV) is the most common type of mutation and is associated with many diseases[1]. Although sequencing approaches can detect SNVs, they do not report on spatial information. Fluorescence in situ hybridization (FISH) can reveal the three-dimensional location of genomic sites of interest through annealing of fluorescently labeled oligonucleotide probes to denatured chromosomal DNA, but generally cannot differentiate highly similar sequences. Recently, advanced RNA FISH methods have been developed to visualize SNVs[2–8]. One of the RNA FISH methods, amp-FISH, has also been implemented to

identify gene loci carrying SNVs by targeting the nascent RNA clusters of expressed genes[9]. However, SNV-sensitive RNA FISH requires the target RNA to be actively transcribing, thereby excluding inactive or very lowly expressed regions. Existing SNV-sensitive DNA FISH requires amplification, either in situ PCR or rolling circle amplification[10–12]. These SNV-sensitive nuclear DNA FISH techniques involve proteinase treatment[11,12], making them incompatible with immunofluorescence imaging of proteins. The vast majority of genome imaging has been performed through direct probe hybridization to target chromatin, for

[1]Department of Biophysics and Biophysical ChemistryJohns Hopkins University School of Medicine, Baltimore, MD 21205, USA. [2]Bloomberg School of Public Health, Johns Hopkins University School of Medicine, Baltimore, MD 21205, USA. [3]Department of Biomedical Engineering, Johns Hopkins University, Baltimore, MD 21205, USA. [4]Department of Urology, James Buchanan Brady Urological Institute, Johns Hopkins University School of Medicine, Baltimore, MD 21205, USA. [5]Department of Oncology, Johns Hopkins University School of Medicine, Baltimore, MD 21205, USA. [6]Sidney Kimmel Comprehensive Cancer Center, Johns Hopkins University School of Medicine, Baltimore, MD 21205, USA. [7]Cellular and Molecular Medicine Graduate Program, Johns Hopkins University School of Medicine, Baltimore, MD 21205, USA. [8]Department of Biophysics, Johns Hopkins University, Baltimore, MD 21218, USA. [9]Howard Hughes Medical Institute, Baltimore, MD 21205, USA. [10]These authors contributed equally: Yanbo Wang, Wayne T. Cottle. ✉e-mail: tjha@jhu.edu

example by probe tiling[13], and to date, direct hybridization DNA FISH with SNV sensitivity has not been realized.

In this work, we develop single guide (sg) version of GOLDFISH (genome oligopaint via local denaturation FISH)[14] to address the technical gap in direct hybridization SNV detection. We show that DNA cleavage efficiency of a specificity-enhancing Cas9 variant in conjunction with a guanine-extended gRNA is sensitive to single base pair differences in vitro and in fixed cells. We further validate sgGOLDFISH probe labeling at three distinct genomic sites, showing high discrimination between point mutations. Finally, we demonstrate allele-specific visualization of disease-associated *LMNA* point mutations in progeroid patient's fibroblasts.

## Results

### Enabling SNV-sensitive GOLDFISH using single guide RNA

The original GOLDFISH method utilized multiple guide RNAs tiling a genomic region of interest in complex with Cas9 nickase (SpCas9 with H840A mutation[15]) to nick genomic DNA at multiple sites in a target region (Supplementary Fig. 1a)[14]. This allowed local denaturation of targeted genomic DNA by loading an engineered helicase, Rep-X[16], to the cleaved strands so that DNA downstream is unwound to expose binding sites for FISH probes (Supplementary Fig. 1a). Because only several kilobases of DNA are unwound, GOLDFISH greatly reduces non-specific binding of FISH probes to other genomic regions compared to conventional FISH that globally denatures genomic DNA. The same GOLDFISH protocol but using catalytically dead Cas9 (dCas9, SpCas9 with D10A and H840A mutations[15]) instead of Cas9 nickase shows minimal labeling[14]. Leveraging the increased sequence-stringency of Cas9 cleavage compared to Cas9 binding, GOLDFISH also has superior signal-to-background ratios compared to methods that rely on Cas9 binding[14].

The use of multiple cut sites in GOLDFISH enabled high efficiency labeling even if the cleavage efficiency at a single site is not very high, but it prevented SNV detection. We hypothesized that, instead of using multiple guide RNAs, GOLDFISH using a single guide RNA (hence called sgGOLDFISH, Supplementary Fig. 1b) may achieve SNV sensitivity if the Cas9 cleavage activity is optimized to be SNV-sensitive (Fig. 1a). Many efforts have been made to improve the cleavage specificity of Cas9 ribonucleoprotein (RNP)[17–20]. For example, the eSp-Cas9(1.1) is an engineered SpCas9 variant that contains three mutations, K848A, K1003A, and R1060A, which has improved DNA cleavage specificity[17]. Extending the 5′ end of a canonical 20-nt protospacer crRNA with an extra guanine (G) in a guide RNA (hereinafter called 5′ extended guide RNA, Supplementary Fig. 2a) also enhances DNA cleavage specificity[21]. Specificity-improved Cas9 variants generally cleave singly mismatched targets as well as their cognate DNA target, but they do not cleave DNA target with two or more mismatches[17–19]. We reasoned that if a mismatch is intentionally introduced to the guide RNA, any additional mismatch on the DNA target, such as an SNV, would abolish cleavage activity of the specificity-improved Cas9 RNP variants. Therefore, a specificity-improved Cas9 variant in complex with guide RNA that carries an intentionally introduced mismatch should be SNV-sensitive. If so, by rationally designing guide RNA and choosing a specificity-improved Cas9 variant, sgGOLDFISH could preferentially label one of the two alleles even when the two alleles differ by only a single nucleotide (Fig. 1a).

We first show that eSpCas9(1.1) in complex with the 5′ extended guide RNA (hereinafter called eCas9 RNP, Supplementary Fig. 2a) can tolerate one mismatch between guide RNA and DNA target, but not two mismatches, demonstrating SNV-sensitivity. We performed in vitro cleavage assays (Supplementary Fig. 2b) to test the cleavage

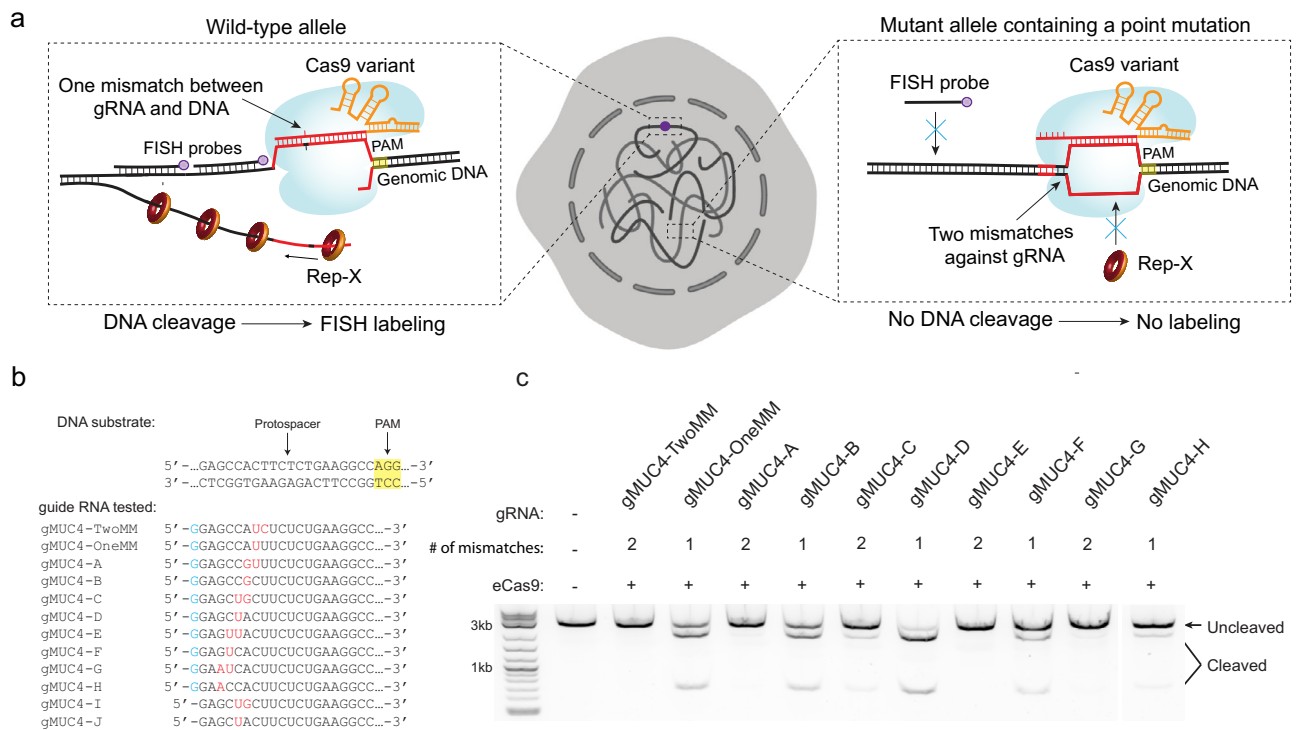

**Fig. 1 | sgGOLDFISH design. a** Schematic of SNV detection using sgGOLDFISH. Genomic DNA in red is homologous to guide RNA. **b** Sequences of DNA substrate and guide RNA tested (only 20 or 21 nucleotides from the 5′ end of the crRNA are shown) in the in vitro cleavage assay. The DNA substrate was PCR-synthesized using human genomic DNA and primers against a non-repetitive region of the *MUC4* gene. A group of guide RNAs with 1 or 2 mismatches against the target protospacer were used in the cleavage assay. The blue "G" represents the 5′ extended guanine of the crRNA. The red colored nucleotides represent mismatches against the DNA substrate. **c** Gel image of the in vitro cleavage assay using guide RNA with 5′ extended guanine. Each DNA cleavage experiment was repeated independently twice with similar results.

activity of eCas9 RNP with guide RNAs bearing different mismatches against their target protospacer (Fig. 1b). We observed efficient cleavage for 4 out of 5 guide RNAs with 1 PAM-distal mismatch, but no cleavage activity for five guide RNAs containing 2 PAM-distal mismatches (Fig. 1b, c). Such specificity was not observed with a canonical 20-nt protospacer guide RNA in complex with eSpCas9(1.1) (Supplementary Fig. 2c). These data indicate that adding an extra PAM-distal mismatch can drastically reduce the cleavage activity of eCas9 RNP.

## SSB-ddPCR for nicking efficiency quantification

Cleaving only the non-target strand of DNA substrate (i.e., the DNA strand that does not base pair with guide RNA) by Cas9 nickase is sufficient for Rep-X unwinding the downstream genomic DNA and FISH probe hybridization in GOLDFISH[14]. We therefore created the eCas9(H840A) variant (hereinafter called eCas9 nickase)[15], and measured its cleavage activity in fixed cells (note that in sgGOLDFISH, cleavage and subsequent steps are performed in fixed cells). The eCas9 nickase cleaves only the non-target strand, resulting in a single-strand break at the target site. Sequencing-based methods have been developed for mapping single-stranded breaks (SSBs) or double-strand breaks (DSBs) in cells, but they are expensive and do not provide the absolute value of the fraction of DNA carrying the break at a target site in the cell population[22–26]. A previously developed droplet digital PCR (ddPCR) assay measures the fraction of DNA with DSBs at a target site, but it is insensitive to SSBs[27]. To quantitatively measure the cleavage efficiency of eCas9 nickase in fixed cells, we modified the ddPCR assay to make it SSB-sensitive (therefore we call the modified ddPCR assay the SSB-ddPCR assay, Fig. 2a). The key modification is that we converted the SSB to DSB by an additional nickase treatment.

In SSB-ddPCR, the eCas9 nickase RNP was applied to fixed and permeabilized HEK293FT cells to cleave its target genomic DNA, which would introduce an SSB at one of the DNA strands if cleavage did occur (Fig. 2a, Step 1, an SSB in the top strand). After harvesting genomic DNA, we used Cas9 nickase to cleave the bottom strand, therefore converting the SSB introduced in Step 1 to a DSB (Fig. 2a, Step 2). If genomic DNA was not cleaved in Step 1, it would become an SSB DNA after Step 2 (Fig. 2a, "If no cleavage" flowchart). The efficiency of the Cas9 nickase to cleave the bottom strand is near 100% under our experimental condition (Supplementary Fig. 3a, b). Therefore, after Step 2, the genomic DNA should be either a DSB DNA (if cleaved in Step 1) or an SSB DNA (if uncleaved in Step 1). Finally, the DNA was mixed with primers and probes for ddPCR (Fig. 2a). The F1/R1 primers span the cleavage sites in Step 1 and Step 2, while the F2/R2 primers do not. The amplification of the F1/R1 and F2/R2 amplicons are detected through FAM and HEX fluorescence, respectively. If a droplet contains a DSB DNA, it is FAM-negative and HEX-positive (Figs. 2b and 2c, green spots), whereas a droplet that is positive for both FAM and HEX contains an SSB DNA (Fig. 2b, c, orange spots). The apparent DSB percentage is estimated by dividing the number of FAM-/HEX + droplets by the number of HEX + droplets (Fig. 2b, c, green percentage numbers). Due to limitations of ddPCR, a genomic DNA not cleaved in Step 1 can still appear as a FAM-/HEX + droplet (Supplementary Note 1). To estimate the true percentage of genomic DNA cleaved by the eCas9 nickase RNP in Step 1 from the SSB-ddPCR readouts, a standard curve was generated from a control experiment (Supplementary Fig. 4 and Supplementary Note 2).

When we performed the SSB-ddPCR experiment using eCas9 nickase in complex with gMUC4-TwoMM, a guide RNA with two mismatches against a target in *MUC4* gene, the apparent DSB percentage

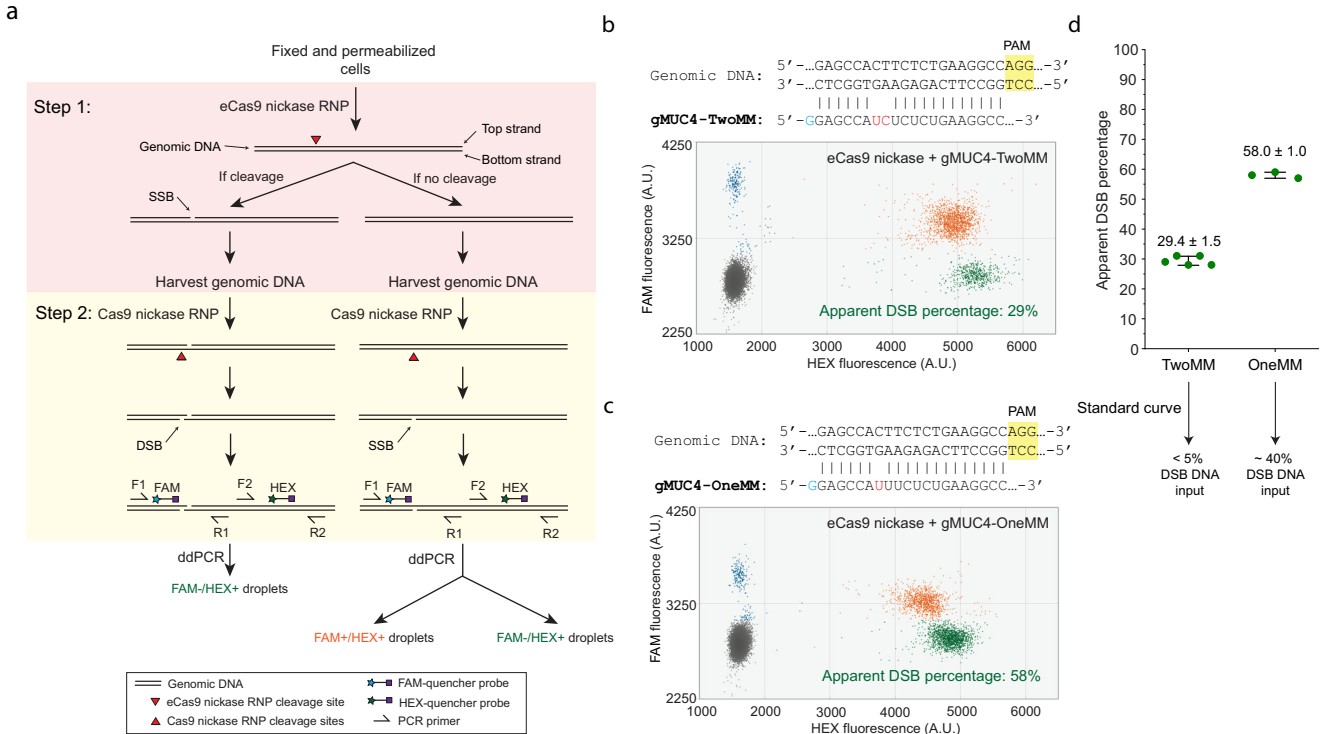

**Fig. 2 | SSB-ddPCR. a** Schematic of SSB-ddPCR. **b** The 5′ sequences of gMUC4-TwoMM, its target genomic DNA sequence, and a representative SSB-ddPCR scattering plot using eCas9 nickase and gMUC4-TwoMM. Apparent DSB percentage of the ddPCR result is shown in green. **c** The 5′ sequences of gMUC4-OneMM, its target genomic DNA sequence, and a representative SSB-ddPCR scattering plot using eCas9 nickase and gMUC4-OneMM. Apparent DSB percentage of the ddPCR result is shown in green. **d** Scatter plot of apparent DSB percentage from the SSB-ddPCR using gMUC4-TwoMM or gMUC4-OneMM. Each dot represents measured apparent DSB percentage from a replicate (*n* = 5 for TwoMM and *n* = 3 for OneMM). Error bar represents mean apparent DSB percentage ± standard deviation, which is also labeled above data points. Their corresponding true DSB percentage input in the ddPCR reactions are indicated below, according to the standard curve in Supplementary Fig. 4b, c. Raw data points underlying each plot are provided as a Source Data file.

was 29.4% ± 1.5% (Fig. 2b, d), corresponding to <5 % target DNA in fixed HEK293FT cells being cleaved by the eCas9 nickase RNP, according to the standard curve (Supplementary Fig. 4b and Supplementary Note 2). However, when using eCas9 nickase with gMUC4-OneMM that carries one mismatch, the apparent DSB percentage was 58.0% ± 1.0% (Fig. 2c, d), corresponding to 40% target DNA cleavage (Supplementary Fig. 4c and Supplementary Note 2). These data demonstrate that an extra mismatch drastically reduced the cleavage efficiency of eCas9 nickase RNP in fixed cells so that a contrast of about an order of magnitude can be obtained.

## Demonstration of sgGOLDFISH

We first tested sgGOLDFISH in proteinase-treated HEK293FT cells using eCas9 nickase with gMUC4-OneMM or gMUC4-TwoMM alongside 23 Cy5-labeled FISH probes against a 1.5-kb non-repetitive region in the *MUC4* gene (*MUC4*-NR) adjacent to the target protospacer (Fig. 3a). Another guide RNA (gMUC4-R) and a Cy3-labeled FISH probe were also designed against a repetitive region (*MUC4*-R) 19-kb from the *MUC4*-NR region to evaluate the specificity and sensitivity of sgGOLDFISH (Fig. 3a). The *MUC4* gene is on the chromosome 3. Although karyotyping of the cell line showed 3 copies of chromosome 3 (one of them is significantly truncated, Supplementary Fig. 5a), we saw majority of cells (72.5%) have only two *MUC4*-R foci (Fig. 3b, green signals, and quantified in Supplementary Fig. 5b). We hypothesized that one of the chromosome 3 lost the DNA fragment containing *MUC4* gene. To test this hypothesis, we imaged another repetitive region (Chr3-repeat) that is 0.27 Mb away from the *MUC4*-R region using a previously reported genome imaging method called CASFISH, which can efficiently label repetitive DNA sequences (Supplementary Fig. 5c). We found 83.8% cells have two Chr3-repeat foci (Supplementary Fig. 5d). These data are consistent with the loss of the *MUC4* gene and the Chr3-repeat region from one of the three chromosomes. Therefore, we expect two copies of *MUC4* gene in each cell. Next, we looked at the labeling efficiency of *MUC4*-NR region by sgGOLDFISH using gMUC4-OneMM or gMUC4-TwoMM. With gMUC4-OneMM, on average 0.56 *MUC4*-NR FISH foci per cell (foci/cell) were detected (Fig. 3b). With gMUC4-TwoMM, on average 0.07 foci/cell were detected (Fig. 3b). With concurrent use of gMUC4-R and gMUC4-OneMM, we found that 52 of 56 *MUC4*-NR foci colocalized with *MUC4*-R foci, indicating high labeling specificity of sgGOLDFISH (Fig. 3c). There were a total of 202 *MUC4*-R foci among which 52 showed colocalization with *MUC4*-NR, suggesting about 26% detection efficiency of sgGOLDFISH for this locus. We then performed sgGOLDFISH without proteinase treatment and observed 0.45 foci/cell for gMUC4-OneMM and 0.03 foci/cell for gMUC4-TwoMM, again demonstrating SNV-sensitivity and suggesting that proteinase treatment is dispensable (Supplementary Fig. 5e).

We next tested the generalizability of sgGOLDFISH by targeting three additional loci. We designed guide RNAs with different number of mismatches against the *ACTB*, *LMNA*, and *CDKN2A* genes (Supplementary Fig. 6a, 6c and 6e) and tested their activities when complexed with eCas9 using the in vitro cleavage assay (Supplementary Fig. 2b). We observed efficient DNA cleavage for fully matched and singly mismatched guide RNA (gACTB, gACTB-OneMM, gLMNA-WT, gCDKN2A, gCDKN2A-OneMM), but not for doubly mismatched guide RNA (gACTB-TwoMM, gLMNA-MUT, gCDKN2A-TwoMM) (Supplementary Fig. 6b, 6d and 6f). We next performed sgGOLDFISH against each of the *ACTB*, *LMNA*, and *CDKN2A* genes (Supplementary Fig. 7a–7c). By performing sgGOLDFISH at the *ACTB* locus (Supplementary Fig. 7a), we observed 0.78 foci/cell for the singly mismatched gACTB-OneMM and 0.02 foci/cell for the doubly mismatched gACTB-TwoMM (Fig. 3d). For the *LMNA* sgGOLDFISH (Supplementary Fig. 7b), we observed 1.17 foci/cell for the singly mismatched gLMNA-WT and 0.09 foci/cell for the doubly mismatched gLMNA-MUT (Fig. 3e). Note that the *LMNA* and *ACTB* genes are on chromosome 1 (three copies)

and chromosome 7 (three copies), respectively (Supplementary Fig. 5a). The full width at half maximum of Gaussian fit to the imaged *LMNA* foci is 617 ± 92 nm (mean ± SD, Supplementary Fig. 7d), indicating sgGOLDFISH yields well-defined FISH spots suitable for subnuclear localization analysis. For the *CDKN2A* sgGOLDFISH (Supplementary Fig. 7c), we observed minimal labeling efficiency even though we used the fully matched guide RNA (Supplementary Fig. 8a and 8b). We hypothesized that the minimal labeling efficiency at this site is because the eCas9 nickase RNP cannot efficiently cleave the *CDKN2A* protospacer in fixed cells. To test this hypothesis, the SSB-ddPCR assay was performed using either eCas9 nickase or dCas9 in complex with the gCDKN2A (Supplementary Fig. 8c). We observed no significant differences between the ddPCR results from eCas9 nickase and dCas9 (Supplementary Fig. 8d and 8e), suggesting that the eCas9 nickase RNP did not cleave the *CDKN2A* protospacer, probably because local environments such as nucleosomes hinder the eCas9 nickase RNP from targeting its protospacer in the fixed cells. Taken together, sgGOLDFISH is generalizable as it worked well in three out of four loci tested, and a single-nucleotide difference can produce about an order of magnitude difference in sgGOLDFISH labeling efficiency.

## sgGOLDFISH for Hutchinson-Gilford progeria syndrome mutation

We next applied sgGOLDFISH in patient-derived Hutchinson-Gilford progeria syndrome (HGPS) fibroblasts (HGADFN167 cells, which have normal diploid karyotype[28]). HGPS cells contain one copy of normal *LMNA* gene (*LMNA*-WT), and one copy of mutated *LMNA* gene (*LMNA*-MUT) that carries a point mutation (c. 1824 C > T) (Fig. 4a), which causes expression of progerin, a truncated gene product, and alterations of nuclear shape[29]. The gLMNA-MUT guide RNA described above contains two mismatches against the wild-type *LMNA* sequence and one mismatch against the progeria mutant sequence, and the gLMNA-WT has one mismatch against the wild-type and two mismatches against the mutant (Supplementary Fig. 9a and 9b). Therefore, sgGOLDFISH using gLMNA-MUT should preferentially label the mutant allele whereas the wild-type allele is preferentially labeled when gLMNA-WT is used (Fig. 1a). To test this prediction, we created HGPS mutation-corrected fibroblasts by delivering adenine base editor ABE7.10max-VRQR (ABE) mRNA and corresponding sgRNA into the HGPS cells[30] (Fig. 4b). ABE is precise, efficient, and convenient in correcting point mutations compared to traditional methods such as helper-dependent adenoviral vectors and Cas9-mediated gene editing[31,32]. Previous studies utilized lentivirus or adeno-associated virus (AAV) to deliver ABE into cells for correcting the HGPS mutation[30,33]. The genome integration sites of lentivirus are generally unpredictable[34]. There are also safety concerns of developing hepatocellular carcinoma in mice after AAV treatment, which is likely due to integration of the AAV genome[30,35]. In contrast, non-viral delivery of the ABE in the form of mRNA does not risk unwanted DNA integration into the genome. Transient expression of the ABE due to the finite lifetime of the delivered mRNA could also reduce potential off-target base editing. Constant expression of ABE may also block the point mutation site as ABE can keep binding at the site even after editing, potentially hindering the binding of our eCas9 nickase RNP to the *LMNA* point mutation site during sgGOLDFISH. Our DNA-free ABE editing efficiently corrected the HGPS mutation (>94% efficiency) (Fig. 4c and Supplementary Fig. 9c). Accordingly, the fraction of morphologically abnormal nuclei was significantly reduced after ABE treatment (Supplementary Fig. 9d and 9e).

To test if sgGOLDFISH preferentially labels the progeria mutant allele with gLMNA-MUT, we made a cell mixture containing 50% uncorrected HGPS cells and 50% ABE-corrected HGPS cells (hereinafter called 1:1 mixture, Supplementary Fig. 9f). sgGOLDFISH against the *LMNA* gene using gLMNA-MUT was applied to the 1:1 mixture in parallel with progerin immunofluorescence, and a cell with at least one

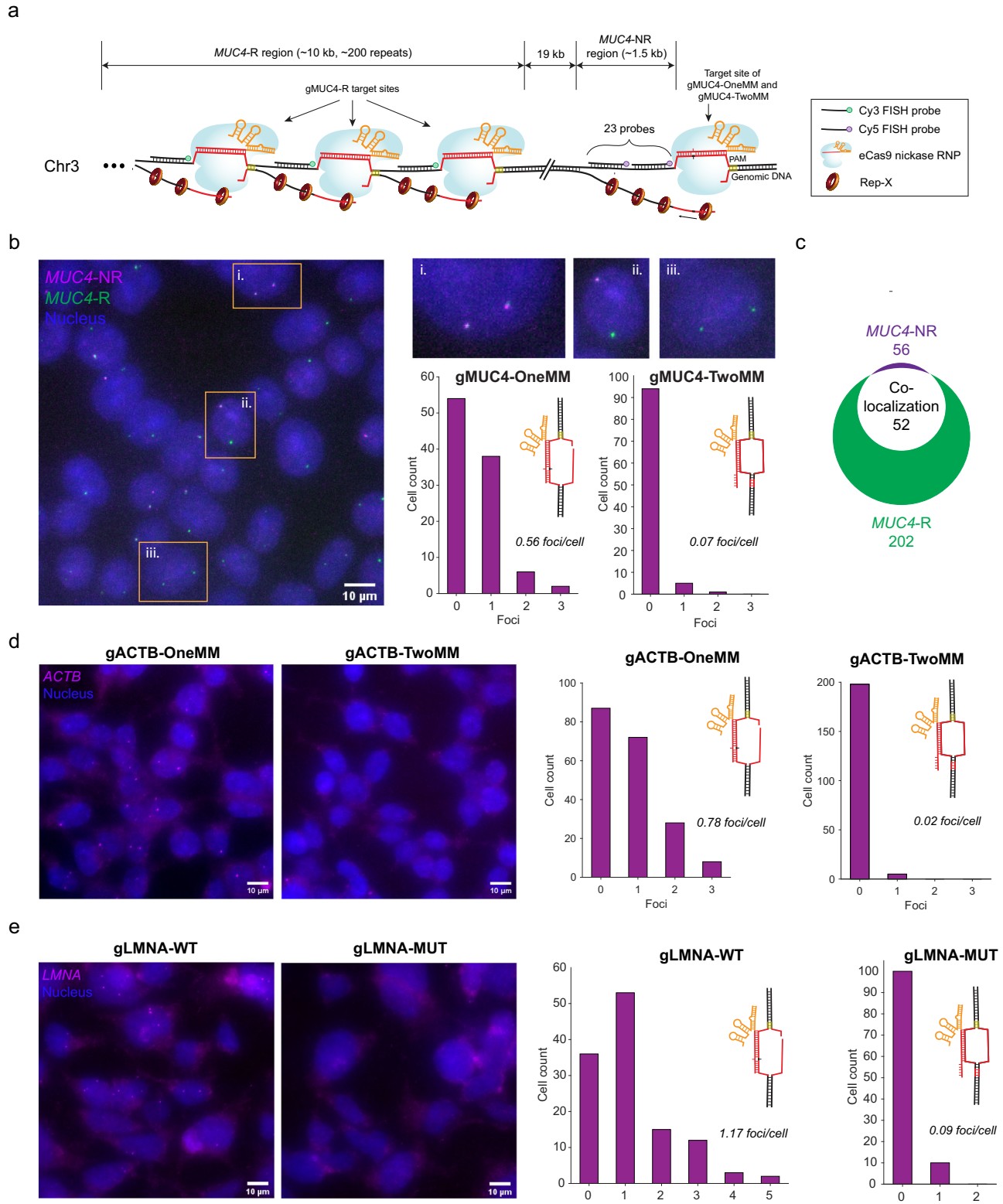

**Fig. 3 | sgGOLDFISH. a** Only one guide RNA (gMUC4-OneMM or gMUC4-TwoMM) and 23 FISH probes were used to target the *MUC4*-NR region. The figure shows the case that gMUC4-OneMM was used (there is one mismatch between guide RNA and target DNA). In contrast, although only one guide RNA (gMUC4-R) and one FISH probe species were used to target *MUC4*-R region, the *MUC4*-R region contains ~200 repeats, therefore multiple binding sites for eCas9 nickase RNP complexed with gMUC4-R and the FISH probe against *MUC4*-R region. **b** A representative sgGOLDFISH image using gMUC4-OneMM. Single cells outlined in orange are magnified on the upper–right corner, which show (i) two detected *MUC4*-NR

alleles, (ii) one detected *MUC4*-NR allele (iii.) no detected *MUC4*-NR allele. Histograms of sgGOLDFISH *MUC4*-NR foci per cell using gMUC4-OneMM (*n* = 78) or gMUC4-TwoMM (*n* = 100). **c** Quantification of co-localized *MUC4*-R and *MUC4*-NR foci. **d** Representative sgGOLDFISH images using gACTB-OneMM and gACTB-TwoMM, and histograms of sgGOLDFISH *ACTB* foci using gACTB-OneMM (*n* = 195) or gACTB-TwoMM (*n* = 203). **e** Representative sgGOLDFISH images using gLMNA-WT and gLMNA-MUT, and histograms of sgGOLDFISH *LMNA* foci using gLMNA-WT (*n* = 121) or gLMNA-MUT (*n* = 110). Raw data points underlying each plot are provided as a Source Data file.

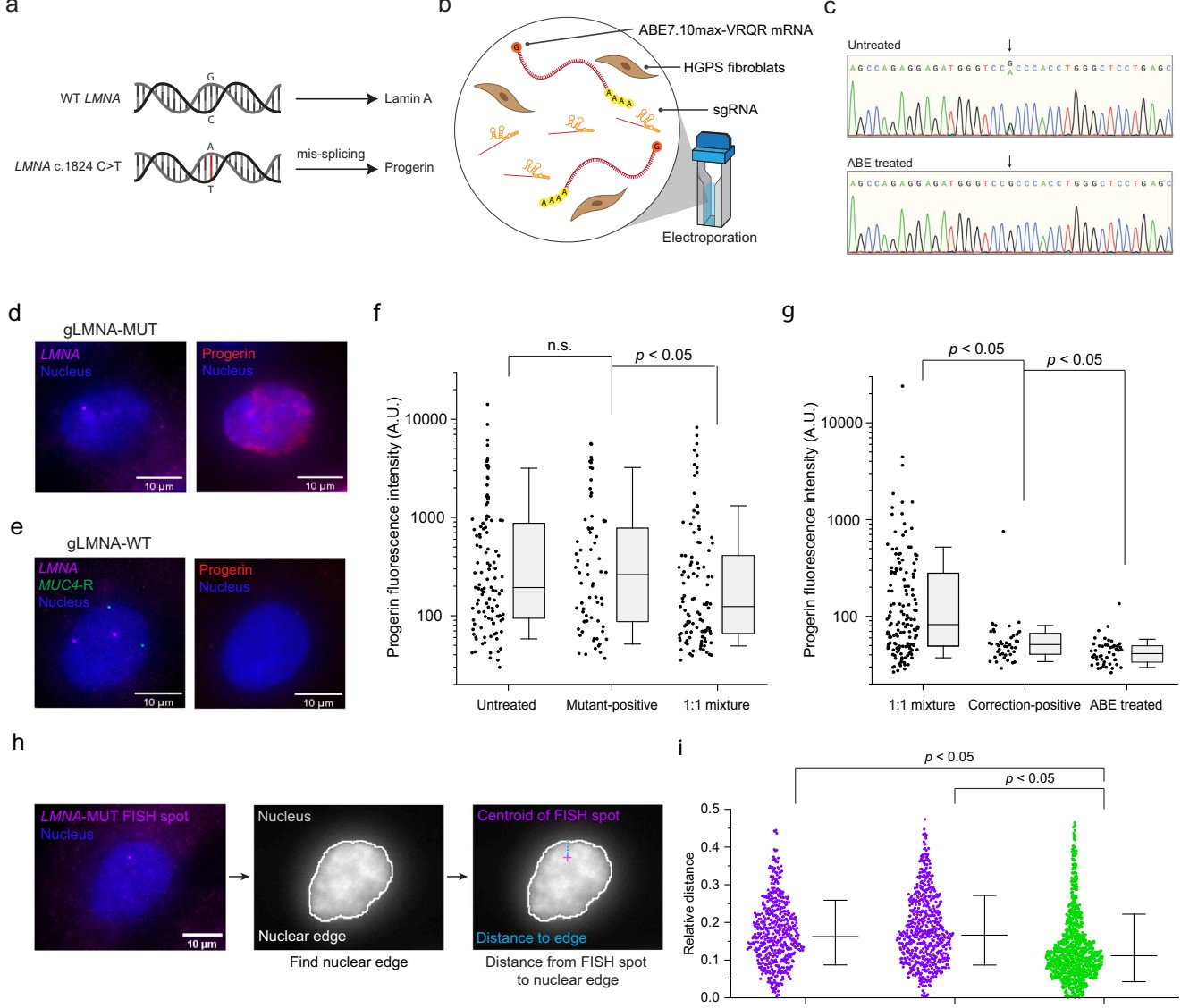

**Fig. 4 | SNV detection in HGPS cells. a** Schematic of HGPS pathogenic point mutation. **b** Schematic of ABE editing of HGPS fibroblasts. **c** Representative Sanger sequencing results of untreated and ABE-treated HGPS fibroblasts. The black arrow indicates the HGPS pathogenic point mutation site. **d** and **e**, sgGOLDFISH in parallel with progerin immunofluorescence using (**d**) gLMNA-MUT or (**e**) gLMNA-WT. **f**, **g** Quantifications of progerin immunofluorescence intensity from different populations. Each dot represents a quantified cell. In Fig. 4f, $n = 113, 69, 107$ for Untreated, Mutant-positive, 1:1 mixture, respectively. In Fig. 4g, $n = 152, 44, 50$ for 1:1 mixture, Correction-positive, ABE treated, respectively. Mann-Whitney $U$ Test (two-sided) is used. n.s. represents $p > 0.05$. The exact $p$ values are 0.91225 (Untreated and Mutant-positive in Fig. 4f), 0.040809 (Mutant-positive and 1:1 mixture in Fig. 4f), 0.000008 (1:1 mixture and Correction-positive in Fig. 4g),

0.001048 (Correction-positive and ABE treated in Fig. 4g). Box represents the range of 25th to 75th percentiles, and whisker represents the range of 10th to 90th percentiles. Median line is shown in the box. A.U., arbitrary units. **h** Schematic of measuring distance from a FISH spot to the nuclear edge. **i** Quantifications of the relative distance (i.e., the distance from a FISH spot to the nuclear edge divided the square root of the nuclear area) of *LMNA*-WT, *LMNA*-MUT and *MUC4* alleles to the nuclear edge. Each dot represents a quantified FISH spot ($n = 538, 636$ and $994$ for *LMNA*-MUT, *LMNA*-WT and *MUC4*, respectively). Student's t test (two-sided) is used. $p = 1.83 \times 10^{-17}$ (*LMNA*-MUT and *MUC4*) and $1.04 \times 10^{-23}$ (*LMNA*-WT and *MUC4*), respectively. Median line is shown. Whisker represents mean ± SD. Raw data points underlying each plot and Sanger sequencing traces are provided as in Source Data files.

*LMNA* sgGOLDFISH spot was assigned as a "mutant-positive cell" (Fig. 4d and Supplementary Fig. 10a). The progerin immuno-fluorescence intensity averaged over the nucleus was comparable between the mutant-positive cells and untreated HGPS cells but was significantly lower for cells randomly selected from the 1:1 mixture (Fig. 4f), consistent with reduced progerin expression after the HGPS mutation correction[30]. Therefore, sgGOLDFISH successfully identified uncorrected HGPS cells from a mixed population.

Next, we performed sgGOLDFISH in the 1:1 mixture again but using gLMNA-WT instead (Fig. 4e and Supplementary Fig. 10b). In parallel, we performed GOLDFISH against the *MUC4*-R region to

simultaneously estimate the cell cycle stage (Fig. 4e, e.g., detection of two *MUC4*-R foci indicates G0/G1). We have previously shown that the GOLDFISH detection efficiency of the *MUC4*-R region in fibroblasts is very high (around 90%)[14]. The *MUC4*-R foci and *LMNA* foci, which should be on chromosome 3 and chromosome 1 respectively, did not co-localize (Supplementary Fig. 10b and 10c), confirming the FISH foci are not non-specific probe aggregations. An ABE-corrected HGPS cell should have 2 to 4 copies of *LMNA*-WT alleles, depending on cell cycle. Therefore, cells with two *LMNA* foci and two *MUC4*-R foci, or four *LMNA* foci and four *MUC4*-R foci, were assigned as "correction-positive cells" (Fig. 4e). The progerin fluorescence was significantly lower for

the correction-positive cells than for cells randomly selected from the 1:1 mixture (Fig. 4g). Taken together, these data suggest that sgGOLDFISH can preferentially label the *LMNA*-WT or the *LMNA*-MUT alleles even though these two alleles differ by only a single base pair, and that we can use sgGOLDFISH to identify base-edited cells or unedited cells from a heterogeneous population. We note that the progerin fluorescence is slightly higher for the correction-positive cells than for ABE-treated HGPS fibroblasts, likely because a small fraction of the *LMNA*-MUT alleles were labeled giving false positives (Fig. 4g).

Direct hybridization of probes allows for accurate localization of sequences of interest. To demonstrate sgGOLDFISH can facilitate sub-nuclear spatial analysis, we measured the distance of *LMNA*-WT, *LMNA*-MUT and *MUC4* foci to the nuclear edge in HGPS fibroblasts. *MUC4* foci appeared closer to the nuclear edge than *LMNA*-WT and *LMNA*-MUT foci (Fig. 4h, i), consistent with Lamin A/C-ChIP (chromatin immuno-precipitation) data, which correlates with a gene's proximity to the nuclear membrane, from the same HGPS line which showed stronger Lamin A/C-ChIP signal for *MUC4* compared to *LMNA* (Supplementary Fig. 10d)[36].

## Discussion

Instead of labeling heterogeneous amplification products as in other DNA FISH methods with SNV sensitivity[11,12] (e.g., the length of rolling circle amplification product varies from 160 kb to 1Mb[37]), sgGOLDFISH labels 1 kb to 1.5 kb chromatin flanking the target SNV. The much smaller labeling region of sgGOLDFISH may explain why proteinase treatment is dispensable for sgGOLDFISH but not for the other methods that require proteinases to digest cellular proteins for the amplification enzymes to function in the nucleus[11,12].

A previous study demonstrated SNV-detection using Cas9 when the target SNV is located in the PAM proximal region (<10 base pairs from PAM)[11]. To fill the gap of PAM-distal SNV detection, here we focused on imaging of SNVs in the PAM-distal region (10 to 20 base pairs from PAM) using sgGOLDFISH. In principle, sgGOLDFISH should be able to detect PAM-proximal SNVs as well. Because Cas9 is more sensitive to PAM-proximal mismatches than PAM-distal mismatches[38,39], SNV-sensitive Cas9 RNP variants should also be able to differentiate DNA targets that differ by a PAM-proximal SNV. Future studies are needed to validate this point.

In this study, the guide RNA of sgGOLDFISH carries an intentionally introduced mismatch at the position immediately next to the intended SNV position. Placing the intentionally introduced mismatch at other positions within the PAM-distal region is likely to work as well because eSpCas9(1.1) has minimal gene editing efficiency at doubly mismatched targets even if two mismatches are not adjacent[17].

The sgGOLDFISH takes advantage of the SNV-sensitive cleavage activity of specificity-improved Cas9 RNP variants. The eCas9 RNP was SNV-sensitive in vitro for the four loci tested in this study (Fig. 1c and Supplementary Fig. 6). However, because the cleavage activity of Cas9 is dependent on local sequence[40], it is possible that the cleavage activity of eCas9 RNP is not optimal for another SNV target, and the design of guide RNA may need to be adjusted. For example, if the protospacer sequence of a new target has higher than 80% GC content, then Cas9 RNP is likely to have low activity on this target[40] and a single mismatch may completely abolish the cleavage activity of Cas9 RNP on this target. In this case, the intentionally introduced mismatch should be removed from the guide RNA. The activity of Cas9 RNP can be fine-tuned by adding/removing/adjusting the intentionally introduced mismatch in the guide RNA. Adding and removing the 5′ extended guanine on the guide RNA can also reduce or increase the cleavage activity[21]. Currently available specificity-improved Cas9 variants other than eSpCas9(1.1) may also work for sgGOLDFISH, adding flexibility for assembling SNV-sensitive Cas9 RNP with optimal activity and specificity against target SNVs[18–20].

Because the necessity of the PAM sequence may put a constraint on SNVs detectable by sgGOLDFISH, we examined 39 prostate cancer-associated single nucleotide polymorphisms (SNPs) that were reported in a recent study[41]. Among the 39 SNPs, 27 SNPs are closed to a potential PAM sequence so that they can be placed within the PAM-distal region of the corresponding protospacer, and 35 SNPs can be placed with the protospacer. This suggests that PAM sequence does not put a significant constraint on the targeting scope of SNV.

The limitations of our method include a requirement to test Cas9 RNP activity for new targets before sgGOLDFISH, slight nuclear shrinkage caused by fixation[14], and ~25% detection efficiency compared to existing SNV-sensitive FISH methods (10%-65%)[2,3,8,10,11]. The detection efficiency of sgGOLDFISH is determined by eCas9 nickase RNP binding and cleavage efficiency, Rep-X unwinding efficiency, and probe hybridization efficiency. In our in vitro cleavage assays (Fig. 1c and Supplementary Fig. 6), adding 50 times excess of the eCas9 RNP in molar ratio cannot completely cleave its DNA substrates, suggesting the intentionally introduced mismatch in guide RNA likely reduces the DNA cleavage efficiency. The local environment such as nucleosomes may also hinder the binding of eCas9 nickase RNP and the DNA unwinding of Rep-X, therefore reduces detection efficiency of sgGOLDFISH. Given that tens of probes tile the target DNA sequence in sgGOLDFISH, probe hybridization efficiency should not be a major limiting factor in sgGOLDFISH. When targeting SNVs in actively expressing genes, SNV-sensitive RNA FISH techniques may also be a good alternative to sgGOLDFISH because even though they have similarly low detection efficiency per target RNA molecule, many copies of the target RNA are present in each cell[2,5,6]. Because sgGOLDFISH probes are designed to bind the genomic regions downstream of both the wild-type and point-mutated sequences, with the eCas9 nickase cleavage differentiating the wild-type and point-mutated sequences, sgGOLDFISH cannot concurrently label the wild-type allele and mutant allele with probes of different colors.

Overall, given the single-nucleotide sensitivity, immuno-fluorescence compatibility, the ability to accurately localize SNVs, and relatively broad SNV targeting scope (see Supplementary Table 1 for comparison with other methods), sgGOLDFISH will be of value to researchers who study point mutation-related diseases or detect precise genome editing such as base editing.

## Methods

### Human Cell Lines

HEK293FT cells were purchased from the American Type Culture Collection and cultured in DMEM with 4.5 g/L glucose, L-glutamine, and sodium pyruvate (Corning, 10-013-CV) supplemented with 10% heat inactivated fetal bovine serum (FBS, Corning 35-011-CV). Hutchinson-Gilford Progeria Syndrome (HGPS) fibroblasts (HGADFN167) were purchased from the Progeria Research Foundation and cultured in high glucose DMEM without L-glutamine (Thermo-Fisher, 11960-440) supplemented with 20% FBS (Corning, 35-011-CV), 1% Penicillin-Streptomycin (ThermoFisher, 15140-122) and 1% Gluta-MAX (ThermoFisher, 35050-061). All cells were maintained at 37 °C in 5% $CO_2$ and imaging dishes were coated with 1 μg/cm² fibronectin then air dried before plating.

### Expression and purification of Cas9 and Rep-X

Cas9 nickase and eCas9 nickase were prepared as described previously with modifications[42]. Cas9 nickase was expressed using the pMJ826 plasmid (addgene, 39316). Mutagenesis was carried out to introduce the H840A mutation into eSpCas9(1.1) variant using pJSC114 plasmid (addgene, 101215) and QuikChange Lightning Site-Directed Mutagenesis Kit (Agilent, 210518). eCas9 nickase was expressed using the mutagenesis-modified pJSC114 plasmid. The plasmids were transformed into NEB BL21(DE3) competent cells (New England Biosciences). Cultures were maintained in Terrific Broth (Invitrogen)

supplemented with 0.4% glycerol at 37 °C until induction at $OD_{600} = 0.5 - 0.6$ at which point the temperature was lowered to 18 °C and cultures were induced with 0.5 mM IPTG (GoldenBio). Pellets were harvested after 16-18 h and resuspended in lysis buffer (50 mM Tris-HCl, 500 mM NaCl, 5% glycerol, 1 tablets per 50 ml protease inhibitor (EDTA-free, Roche), 0.2 mM PMSF, 1 mM TCEP, 1 mg/ml lysozyme, pH 7.5) and sonicated at 30% amplitude with 2 s on, 4 s duty cycle for 2 min, 3 times. Lysate was spun down and supernatant was mixed with 2 ml Ni-NTA resin (Qiagen) per 50 ml sample and incubated for 1 h at 4 °C, then spun down and decanted. Resin was incubated with Wash Buffer (50 mM Tris-HCl, 500 mM NaCl, 10 mM imidazole, 5% glycerol, 1 mM TCEP, pH 7.5) at 4 °C for 5 min repeated 4 times then added to gravity column. Colum was then incubated with Elution Buffer (50 mM Tris-HCl, 500 mM NaCl, 1 mM TCEP, 300 mM imidazole, 5% glycerol, pH 8–8.5) and fractions were analyzed via denaturing PAGE. Samples were then desalted and concentrated using a 40kD cut off filter into storage buffer (300 mM NaCl, 10 mM Tris-HCl, 0.1 mM EDTA, 50% glycerol, pH 7.5). Ni-NTA purification with desalting showed sufficient purity and activity for GOLD FISH applications and did not require further size selection chromatography.

Rep-X was prepared the same as previously described[14]. pET28a(+) with *rep* (C18L/C43S/C167V/C612A/S400C) was transformed into *E. coli* B21(DE3) (Sigma-Aldrich, CMC0014). A single colony was picked and grown in TB at 37 °C overnight, followed by 30 °C overnight. When OD reached the range between 0.3 and 0.4, the cells were moved to an 18 °C incubator. When OD reaches 0.6 to 0.8, the cells were induced expression with 0.5 mM IPTG and continue growth overnight. The cells were harvested by centrifugation for 15 min at 10,700 × g and 4 °C. The pellet was resuspended in 40 ml of the lysis buffer (50 mM Tris-HCl pH 7.5, 5 mM Imidazole, 200 mM NaCl, 20% (w/v) sucrose, 15% (v/v) glycerol, 17.5 µg/ml PMSF, and 0.2 mg/ml Lysozyme) and sonicate to lyse the cells. The lysed cell mix was centrifuged at 30,000 × g at 4 °C for 30-60 min and collect the supernatant. The supernatant was stir-mixed with pre-equilibrated Ni-NTA resin for 1.5 h at 4 °C. Ni-NTA purification was performed by washing the protein-bound resin with buffer A (50 mM Tris-HCl pH 7.5, 5 mM Imidazole, 150 mM NaCl, 25% (v/v) glycerol), followed by buffer A1M (50 mM Tris-HCl pH 7.5, 5 mM Imidazole, 1 M NaCl, 25% (v/v) glycerol) to remove any DNA residue, and final washed the protein-bound resin with buffer A, then eluted the Rep variant with imidazole buffer (50 mM Tris-HCl pH 7.5, 205 mM Imidazole, 150 mM NaCl, 25% (v/v) glycerol). 20 µM eluted Rep variant was mixed with 100 µM BMOE crosslinker to self-crosslink into Rep-X. The reaction was stir-mixed at room temperature for 1 h. The excess crosslinker and Imidazole was removed by an overnight dialysis and stored in Rep-X storage buffer (50% glycerol, 600 mM NaCl, 50 mM Tris-HCl pH 7.5) at −80 °C.

## Genome sequences
The GRCh38.p13 Primary Assembly was used in this study and downloaded from NCBI. The coordinates of target loci are listed below:

*MUC4*-R region (Chr3: 195788656-195778790)
*MUC4*-NR region (Chr3: 195807684-195808777)
*ACTB* region (Ch7: 5570593-5572592)
*LMNA* region (Chr1: 156137082-156138607)
*CDKN2A* region (Ch9: 21971000-21972999)

## sgGOLDFISH guide RNA and probe design
The SNV site should be within a protospacer of SpCas9. Because the previous study has demonstrated to image SNV at the PAM-proximal region[11], here we focused on testing SNVs located at PAM-distal region. The 13rd to 18th positions from the PAM are ideal (Fig. 1c). Because the eCas9 RNP can tolerate one PAM-distal mismatch, but two PAM-distal mismatches essentially inhibit cleavage under our conditions (Fig. 1c and Supplementary Fig. 6), an additional mismatch was intentionally introduced into the guide RNA (e.g., the U at the 8th position from the 5′

end of crRNA in gMUC4-TwoMM and gMUC4-OneMM, Fig. 1b). Oligo FISH probes for sgGOLDFISH were designed using Oligoarray[43]. The target DNA sequence (~1.5 kb) immediately following the target protospacer was input into the Oligoarray 2.1 with the following constraints: Length: 18- to 24-nt; Tm: 70 °C to 90 °C; %GC: 30-70; Max. Tm for structure: 54 °C; Min. Tm to consider X-hybrid: 54 °C; and there was no consecutive repeat of 5 or more identical nucleotides. Probes that can non-specifically bind to human genome, human noncoding RNA and *E.coli* tRNA were removed. The probe sequences are listed in Supplementary Data 1.

## Preparation of DNA and RNA
The designed oligo FISH probes (without any labeling/modification) were purchased from IDT, and fluorescently labeled as previously described[44]. Briefly, to conjugate an amino-ddUTP at the 3′ end of each oligonucleotide, 66.7 µM DNA oligonucleotides, 200 µM Amino-11-ddUTP (Lumiprobe) and 0.4U/µl Terminal Deoxynucleotidyl Transferase (TdT, Thermo Scientific, EP0162) were mixed in 1X TdT Reaction buffer (Thermo Scientific) and incubated overnight at 37 °C. The reaction was cleaned up by ethanol precipitations and P4 beads (Bio-Rad, #1504124) spin columns. Next, the DNA oligonucleotides conjugated with amino-ddUTP were mixed with 100 µg of Cy3-NHS or Cy5-NHS (Lumiprobe or GE Healthcare) in 0.1 M sodium bicarbonate and incubated overnight at room temperature, and cleaned up by ethanol precipitations and P4 beads (Bio-Rad, #1504124) spin columns. Unlabeled oligonucleotides were removed by high-performance liquid chromatography (HPLC). The DNA substrates for in vitro cleavage assays were synthesized using Phusion® Hot Start Flex 2X Master Mix (NEB, M0536S) and purified using GeneJET PCR Purification Kit (Thermo Scientific, K0701). The primers were purchase from IDT and sequences are listed in Supplementary Data 1. For the guide RNA against the *MUC4* gene, crRNA was synthesized in vitro using HiScribe™ T7 Quick High Yield RNA Synthesis Kit (NEB, E2050S) according to the manufacturer's instructions, and purified by polyacrylamide gel electrophoresis. Alt-R® CRISPR-Cas9 tracrRNA (IDT) was purchase from IDT. The guide RNA was annealed by mixing crRNA and tracrRNA at 1:1 ratio in Nuclease Free Duplex Buffer (IDT), and incubating at 95 °C for 30 s, then slowly cooling down to room temperature over 1 h. For other guide RNAs used in this study, the guide RNA was synthesized using EnGen® sgRNA Synthesis Kit, *S. pyogenes* (NEB, E3322V) according to the manufacturer's instructions. The template DNA sequences are listed in Supplementary Data 1.

## In vitro cleavage assay
For Fig. 1c, Supplementary Fig. 2c and Supplementary Fig. 6, Cas9 RNP was assembled by mixing 200 nM eCas9 nickase with 400 nM guide RNA in the cleavage buffer (20 mM Hepes pH 7.5, 100 mM KCl, 7 mM MgCl₂, 5% (v/v) glycerol and 0.1% (v/v) TWEEN-20, freshly added 1 mM DTT), and incubated for 10 min at room temperature. Then 4 nM DNA substrate was added, and incubated at 37 °C for 1 h (Supplementary Fig. 2b). Next, 80 unites/mL of proteinase K (NEB, P8107S) was added to the reaction, and incubated at 37 °C for 30 min. The reaction was directly loaded into the agarose gel for electrophoresis. For Supplementary Fig. 3b, 400 nM Cas9 nickase RNP cleaving the top strand and 400 nM Cas9 nickase RNP cleaving the bottom strand (the same Cas9 nickase RNP as used in Step 2 in Fig. 2a) were assembled by mixing Cas9 nickase and corresponding guide RNA at 1:1 ratio and incubated for 10 min at room temperature. Then, 600 ng PCR-synthesized DNA substrate was added to the mixture and incubated for 1 h at 37 °C (Supplementary Fig. 3a). Next, 80 unites/mL of proteinase K (NEB, P8107S) was added to the reaction, and incubated at 37 °C for 30 min. The reaction was heated at 90 °C for 1 min to dissociate the two parts of the double-nicked DNA, followed by agarose gel electrophoresis. DNA ladder (NEB, N3200) was used as molecular weight markers. Supplementary Fig. 3b was performed using 0.8% agarose gel and

imaged using Azure Biosystems c150 gel imager. Other gel electrophoreses were performed using 2% agarose gel and imaged using Amersham Typhoon (Cytiva). Uncropped gel images (the whole field of views from the gel imager when imaging the gels) were included in the Source Data.

## Cell fixation

The HEK293FT or HGPS cells adhered to the glass surface of an imaging dish were fixed at −20 °C for 15 min in pre-chilled MAA solution (methanol and acetic acid mixed at 1:1 ratio), then washed three times (5 min each wash at room temperature unless indicated) with PBS. For SSB-ddPCR and the sgGOLDFISH in Fig. 3b, an additional protease treatment was performed: 0.1% pepsin in 0.1 M HCl was applied to the fixed HEK293FT cells and incubated for 45 s at 37 °C, and the cells were washed with PBS once, and incubated in 70%, 90% and 100% EtOH at room temperature, each for 1 min. The cells were then washed three times with PBS.

## SSB-ddPCR

The SSB-ddPCR was performed similarly to DSB-ddPCR[27] with modifications (Fig. 2a). The fixed and pepsin treated HEK293FT cells adhered to the glass surface of the imaging dish were incubated in the binding-blocking buffer (20 mM Hepes pH 7.5, 100 mM KCl, 7 mM MgCl$_2$, 5% (v/v) glycerol and 0.1% (v/v) TWEEN-20, 1% (w/v) BSA, freshly added 1 mM DTT, freshly added 0.1 mg/ml *E.coli* tRNA) for 10 min at 37 °C. Next, 100 nM eCas9 nickase was mixed with 200 nM gMUC4-TwoMM or gMUC4-OneMM in the binding-blocking buffer, and incubated for 10 min at room temperature. The 100 nM eCas9 nickase RNP was then applied to the cells, and incubated for 45 min at 37 °C. After that, 2 mM ATP was supplied to the 100 nM eCas9 nickase RNP solution (i.e., the 100 nM eCas9 nickase RNP in the binding-blocking buffer supplied with 2 mM ATP), and incubated the cells in the solution for another 90 min at 37 °C, followed by PBS wash 3 times. To harvest genomic DNA from the cells, 60 mAU/mL proteinase K (Qiagen, 69504) diluted in PBS was applied to the cells and incubated for 30 min at 37 °C. The solution was collected from the imaging dish, and genomic DNA was extracted using the DNeasy Blood & Tissue Kits (Qiagen, 69504) by following manufacturer's protocol. The extracted genomic DNA (less than 8 ng/μL) was further treated with 400 nM Cas9 nickase RNP using the corresponding guide RNA in 1X NEBuffer r3.1 (NEB, B7203S) for 1 h at 37 °C. Next, 45 unit/mL proteinase K (NEB, P8107S) was added to the reaction and incubated for 30 min at 37 °C. The genomic DNA was purified using Genomic DNA Clean & Concentrator-10 (Zymo, D4011) and eluted in water. Finally, 20 to 50 ng the genomic DNA was mixed with 250 nM probes, 900 nM primers and 250 unit/mL EaeI (NEB, R0508S) in 1X ddPCR Supermix for Probes (no dUTP) (Bio-Rad, 1863023). Droplets were created using Droplet Generation Oil for Probes, DG8 Gaskets, DG8 Cartridges, and QX200 Droplet Generator (Bio-Rad); Droplets were transferred to a 96-well PCR plate and heat-sealed using PX1 PCR Plate Sealer (Bio-Rad). PCR amplification was performed with the following conditions: 95 °C for 10 min, 40 cycles of (94 °C for 30 s, 55 °C for 30 s, 72 °C for 2 min), 98 °C for 10 min, 12 °C hold. Droplets were then individually scanned using the QX200 Droplet Digital PCR system (Bio-Rad). To generate the standard curve, the complex of gMUC4-OneMM and dCas9 (instead of eCas9 nickase) was applied to the fixed and pepsin treated HEK293FT cells as described above, and the genomic DNA was harvested (Supplementary Fig. 4a, Step 1). Half of the genomic DNA was treated with Cas9 nickase RNP as described above, which produces SSB DNA. Another half of the genomic DNA (less than 8 ng/μL) was treated with 0.2 unit/μL MseI (NEB, R0525S) for 1 h at 37 °C, and MseI was deactivated by incubating the reaction 20 min at 65 °C. Following the deactivation, the MseI-treated genomic DNA was purified using Genomic DNA Clean & Concentrator-10 (Zymo, D4011) and eluted in water, which produces DSB DNA. The SSB DNA and DSB DNA were

then mixed at different ratios (0% DSB DNA, 5% DSB DNA,…, 100% DSB DNA), and performed ddPCR using the same settings describe above. SSB-ddPCR against the *CDKN2A* gene was conducted as above with Nb.BbvCI (R0631S, NEB) nicking the bottom strand, an annealing temperature of 56 °C and HEX probes against the cut region while FAM probes against the control region.

## sgGOLDFISH

The cells adhered to the glass surface of an imaging dish were incubated in the binding-blocking buffer (20 mM Hepes pH 7.5, 100 mM KCl, 7 mM MgCl$_2$, 5% (v/v) glycerol and 0.1% (v/v) TWEEN-20, 1% (w/v) BSA, freshly added 1 mM DTT, freshly added 0.1 mg/ml *E.coli* tRNA) for 10 min at 37 °C. Next, 100 nM eCas9 nickase was mixed with 200 nM guide RNA in the binding-blocking buffer, and incubated for 10 min at room temperature. For concurrent *MUC4*-R GOLDFISH (Fig. 3b), additional 20 nM eCas9 nickase and 40 nM gMUC4-R were also assembled in the binding-blocking buffer. After that, 2 mM ATP and 300 μM Rep-X was supplied to the 100 nM eCas9 nickase RNP solution (i.e., the 100 nM eCas9 nickase RNP in the binding-blocking buffer supplied with 2 mM ATP and 300 μM Rep-X), and incubated the cells in the solution for another 90 min at 37 °C, followed by PBS wash 3 times. Next, RNase Cocktail™ Enzyme Mix (Invitrogen, AM2286) was diluted 100 times in PBS and incubated with the cells for 1 h at 37 °C. The cells were washed three times with PBS. The cells were then incubated in freshly made hybridization buffer (20% (v/v) formamide, 2X saline-sodium citrate (SSC), 0.1 mg/mL *E.coli* tRNA, 10% (w/v) dextran sulfate, 2 mg/mL BSA) for 10 min at room temperature. Fluorescently labeled oligo FISH probes (1 nM for *MUC4*-R, 2.5 nM per oligo FISH probe for *MUC4*-NR and *LMNA*, i.e., 57.5 nM and 90 nM final probe concentration for *MUC4*-NR and *LMNA*) in the hybridization buffer were applied to the cells and incubated for 1 h at room temperature (repetitive targets) or 37 °C (non-repetitive targets). The cells were washed twice (10 min each wash) with wash buffer (25% formamide, 2X SSC) at 37 °C and once with PBS at room temperature for 5 min. One drop of Hoechst 33342 Ready Flow™ Reagent (Invitrogen, R37165) was mixed with 2 ml of PBS and incubated with the cells for 2 min at room temperature. Finally, imaging buffer (2X SSC and saturated Trolox (>5 mM), 0.8% (w/v) dextrose) supplied with GLOXY (1 mg/mL glucose oxidase, 0.04 mg/mL catalase) was added to the cells for imaging.

## Immunofluorescence

For progerin immunofluorescence, the cells after sgGOLDFISH were incubated in IF buffer (1X Blocker™ BSA in PBS (Thermo Scientific, 37525) supplied with 0.1% Tween-20) at room temperature for 20 min. Progerin Monoclonal Antibody (13A4) (Thermo Scientific, 39966) was diluted 500 times in the IF buffer, and applied to the cells for overnight incubation at 4 °C. The cells were washed three times with PBS, and incubated with 500 times diluted Alexa750-labeled secondary antibody (Invitrogen, A-21037) in the IF buffer for 30 min at room temperature. Finally, the cells were washed 3 times with PBS and imaged in the imaging buffer. For Lamin A/C immunofluorescence, the cells after fixation were incubated in IF buffer (1X Blocker™ BSA in PBS (Thermo Scientific, 37525) supplied with 0.1% Tween-20) at room temperature for 20 min. Anti-Lamin A + Lamin C antibody [4C11] (Abcam, ab238303) was diluted 500 times in the IF buffer, and incubated with the cells for 1 h at room temperature. The cells were washed three times with PBS, and incubated with 500 times diluted Alexa750-labeled secondary antibody (Invitrogen, A-21037) in the IF buffer for 30 min at room temperature. Finally, the cells were washed 3 times with PBS and imaged in the imaging buffer.

## Chr3-repeat CASFISH

Cas9 imaging of Chr3 copy number was accomplished using a Cas9 RNP complex containing Cy3 labeled crRNA complementary to the Chr3 repetitive site. Cells were fixed using 1:1 methanol:acetic acid at

−20 °C in imaging dishes. Dishes were blocked using 1× Blocker™ BSA supplemented with 0.1% tween-20 at 37 C for 30 min then Cy3-Cas9 RNP was added at 5 nM in Cas9 reaction buffer (20 mM HEPES pH 7.5, 100 mM KCl, 1 mM MgCl$_2$, 5% v/v glycerol, 0.1% tween-20, 1% BSA, and freshly added 1 mM DTT) for 1 h at 37 °C. Dishes were then washed with Cas9 reaction buffer 3x for 5 min and nuclei were stain with Hoechst 33258 (1 µg/ml) for 10 min at room temperature. Finally, imaging buffer supplied with GLOXY was added to cells for imaging.

## Microscopy

sgGOLDFISH imaging was performed using Nikon Eclipse Ti microscope equipped with Nikon perfect focus system, Xenon arc lamp. The system was driven by Elements software. Nikon 60X/1.49 NA objective (CFI Apo TIRF) was used. Emission was collected using a custom laser-blocking notch filter (ZET488/543/638/750 M) from Chroma. Images were recorded using an electron-multiplying charge-coupled device (Andor iXon 888). Images were recorded as z-stacks (20 to 30 steps), with 300 nm to 500 nm step size.

## DNA-free base editing in HGPS fibroblasts

The guide RNA for base editor to correct the HGPS mutations (LMNA-VRQRABE-sgRNA) and DNA primers for preparing VRQR-ABE7.10max mRNA were purchased from IDT (see Supplementary Data 1 for the RNA and DNA sequences). To prepare DNA template for VRQR-ABE7.10max mRNA, pUC19 (NEB, N3041S) was linearized using EcoRI-HF (NEB, R3101S) and HindIII-HF (NEB, R3104S). VRQR-AEB fragment was PCR-synthesized using the Plasmid #154429 (addgene) and VRQR-AEB-primer-F and VRQR-AEB-primer-R, and gel purification was carried out to remove non-specific products. Mutagenesis was performed using pcDNA3.3-eGFP (addgene, Plasmid #26822) and T7-Mutagenesis-primer-F and T7-Mutagenesis-primer-R to replace the "G" following the T7 promoter sequence with an "A" by the QuikChange Lightning Site-Directed Mutagenesis Kit (Agilent, 210518). T7-5′UTR fragment was PCR-synthesized using the mutagenesis-modified pcDNA3.3-eGFP and T7-5′UTR-primer-F and T7-5′UTR-primer-R. The 3′UTR fragment was PCR-synthesized using the mutagenesis-modified pcDNA3.3-eGFP and 3′UTR-primer-F and 3′UTR-primer-R. Next, the linearized pUC19, VRQR-AEB fragment, T7-5′UTR fragment, 3′UTR fragment was assembled into a plasmid (VRQRABE-mRNA plasmid) using NEBuilder HiFi DNA Assembly Master Mix (NEB, E2621S) according to manufacturer's protocol. The linear VRQRABE-mRNA DNA template was PCR-synthesized using VRQRABE-mRNA plasmid, VRQRABE-mRNA-linearTemplate-F and VRQRABE-mRNA-linearTemplate-R. All PCR reactions were performed using Q5® Hot Start High-Fidelity 2X Master Mix (NEB, M0494S). The in vitro transcription of VRQR-ABE7.10max mRNA reaction contains 50 ng/µL linearized VRQRABE-mRNA DNA template, ATP/CTP/GTP (5 mM each), 5 mM N1-methylpseudouridine (TriLink, N-1081-1), 4 mM CleanCap AG (TriLink, N-7113), 1 unit/µL Murine RNase Inhibitor (NEB, M0314S), 0.002 units/µL Yeast Inorganic Pyrophosphatase (NEB, M2403S) and 8 units/µL T7 RNA Polymerase (NEB, M0251S) in the transcription buffer (40 mM Tris-HCl pH 8, 20 mM spermidine, 0.02% (v/v) Triton X-100, 165 mM magnesium acetate, freshly added 10 mM DTT). The in vitro transcription reaction was incubated at 37 °C for 2 h, and treated with DNase I by supplying with 1X DNase buffer (NEB, B0303S) and 0.3 units/µL DNase I (M0303S) and incubating at 37 °C for 20 min. The reaction was purified using Megaclear™ Transcription Clean-Up Kit (Invitrogen, AM1908), and dephosphorylated using 0.25 units/µL Antarctic Phosphatase (NEB, M0289S) according to manufacturer's protocol. The VRQR-ABE7.10max mRNA was purified again using the Megaclear™ Transcription Clean-Up Kit. All electroporation experiments were carried out using the Lonza 4D-Nucleofector System. For mRNA editing in HGPS cells, 5 µg of LMNA-VRQRABE-sgRNA was mixed in a total 25 µL volume (SE kit, Lonza) then resuspended with 200k HGPS cells and electroporated using them with CM-120 setting. Cells were maintained at 37 °C in 5% CO$_2$ for 3 days before collecting genomic DNA using DNeasy Blood & Tissue Kits (Qiagen, 69504) and sequencing.

## Data analysis for GOLD FISH

Images were processed using Fiji/ImageJ. Z-stack images were projected to a single plane using the 'Max Intensity' Z-Projection function. The contrasts of images were linearly adjusted by changing the minimum and maximum values using the 'brightness/contrast' function in ImageJ for optimal visualization purpose only. FISH-quant was used to find foci in each cell and fitted with three-dimensional (3D) Gaussian function[45]. To measure Progerin immunofluorescence intensity, 'Subtract Background' function (radius: 150 pixels) in ImageJ was applied to images so that the image area without sample has minimal intensity compared to the image area with cells. The nuclear edge, nuclear area and the distance from a FISH focus to the nuclear edge were analyzed using custom-written MATLAB scripts. OriginPro 2021b was used for generating plots.

## Statistics & Reproducibility

No statistical method was used to predetermine sample size. We aimed at imaging many cells in different field of views (cell numbers imaged are indicated in figure legends), which we empirically found to be sufficient to obtain reproducible results. Cells with overlapped nuclei (i.e., cells are too close to each other) are excluded from quantification of FISH foci/cell because the quantification is for single cells. The study did not involve the treatment of human subjects or laboratory animals, so the experiments were not randomized. The investigators were not blinded to allocation during experiments and outcome assessment. Each experiment was repeated independently twice with similar results.

## Reporting summary

Further information on research design is available in the Nature Portfolio Reporting Summary linked to this article.

## Data availability

The data generated in this study (e.g., z-stack raw images) have been deposited in at Mendeley data (https://doi.org/10.17632/mh9mzgz2nn.1). Source data are provided with this paper. The GRCh38.p13 Primary Assembly was used in this study and downloaded from NCBI (https://www.ncbi.nlm.nih.gov/assembly/GCF_000001405.39/). The Lamin A/C-ChIP data of the HGPS fibroblasts was downloaded from NHGRI (https://research.nhgri.nih.gov/manuscripts/Collins/HGPSepigenetics/download.shtml). Source data are provided with this paper.

## Code availability

The custom-written MATLAB scripts used in the study are available on Github (https://github.com/ywang285/NuclearEdge).

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

## Acknowledgements

We thank the Doudna laboratory (University of California-Berkeley) for generously providing dCas9 stocks. We thank the Regot lab (Johns Hopkins School of Medicine) for generously providing HEK293ft cell line. We thank the support of Worod Allak (Johns Hopkins School of Public Health) on ddPCR experiments. We acknowledge support from The Progeria Research Foundation. The project was supported by grants from the National Institutes of Health (GM122569 to T.H. and GM097330 to S.B) and the National Science Foundation [PHY-1430124 to T.H.]. W.T.C. was supported by the NIH training grant T32 GM007445 and NSF GRFP. T.H. is an investigator with the Howard Hughes Medical Institute.

## Author contributions

Y.W., W.T.C., and T.H. designed the experiments. Y.W. and W.T.C. performed sgGOLDFISH and ddPCR experiments. M.T.P and S.Y. assisted with sgGOLDFISH target selection. R.Z. assisted with ddPCR experiments. Y.W. synthesized ABE mRNA. W.T.C. cultured cells and performed electroporation. H.W. synthesized and purified crRNA. M.G. and W.T.C. expressed and purified proteins. Y.W. and W.T.C. performed data analysis. Y.W., T.H., and W.T.C. discussed the data. Y.W., T.H., W.T.C., and S.B. wrote the manuscript.

## Competing interests

The authors declare no competing interests.
