## [Peer Review File · Nature Communications]

Reviewers' Comments:

Reviewer #1:

Remarks to the Author:

In this manuscript Taekjip Ha and colleagues describe a variant of their technique called GOLDFISH that was described earlier in Wang, Y. et al. *Molecular Cell* (2021). In the present manuscript they apply it for point mutation detection. In this technique a Cas9-nickase is used to bind to the perfect target but not to a mismatched target. The nicked strand is displaced by a helicase and the exposed target strand is then detected using smFISH probes. In order to achieve the required specificity, the guide RNAs are designed so that they contain an additional mismatch against the wild-type (and mutant) sequence as a result, the mutant target is mismatched at one position and the wild-type is mismatched at two. Combined specificity of Cas9 cleavage with smFISH is a great strength. Convincing results are shown for two SNPs. The paper is well-written and the quality of the work appears to be high. Here are some concerns (not in the order of significance):

1. Efficiency of detection is 26% which is disappointing. This may severely impact the utility of the technique, especially when more than one SNP will need to be detected from the same cell. Imagine you need to detect two SNP's in the same cell, if the probability of detection of one is 26%, the probability of detection of both will be only 7%. Also, if you are imaging a tissue section for just one SNP, 75% of the cells will be missed. This is not the case with some of the competing technologies, such as ampFISH (Marras et al *PNAS*, 2019), which even though has similarly low detection efficiency per target mRNA molecule, it will classify the cell correctly because many mRNA molecules are present in each cell.
2. Related to this, what do the authors think is the reason for low efficiency detection? Is Cas9 not binding because of the additional mismatch, or is the Cas9 is inherently inefficient? The use of a repeated sequence as a standard to locate all copies the MUC4 locus has worked out well in this system, but that is not a general case.
3. In order to evaluate the performance of the second system, LMNA locus, the author should provide its copy number status, does it have two copies or more?
4. A possible weakness is that only one variant is reported by the system. The competing technology, ampFISH reports wild-type in one color and the mutant in the second cone. Therefore wild-type, mutant and heterozygotes can be distinguished for any expressing gene locus.
5. Another possible concern is how general is this technique given that the Cas9 system is rather tolerant of mismatches, even more than one (<https://www.nature.com/articles/nbt.2675>). How will the design for a new SNP work? Where would one place the deliberate mismatch in relation to the intended SNP, and where would the intended SNP be located? Also, doesn't the necessity of the PAM sequence put a significant constrain on what SNP's are detectable?
6. In order to show generality of the approach, I suggest inclusion of several other unique SNP's if they can be included without significant new work.
7. The point about bulky amplification products in other methods is not so strong, given that the length of rolling circle product is controllable by the time of incubation. In fact, for the routine implementation of the sgGOLDFISH in tissue sections, it will be necessary to have brighter foci than produced by 23 probes and Rolling circle amplification will do better. Also, please site the paper where the idea of probe-tiling was first described.
8. It is great that they included a comparison with other techniques in supplementary table 1. However, it doesn't include more modern techniques like ampFISH, which was designed for RNA but has been implemented for gene locus identification for expressing genes (Fraser et al *PNAS* 2021). By saying "SNV-sensitive RNA FISH requires the target RNA to be actively transcribing, thereby excluding nongenic regions and inactive or stochastically expressed genes" early in discussion they are bypassing a comparison with RNA detection techniques.

Reviewer #2:

Remarks to the Author:

The authors described an in situ genomic imaging approach with single-nucleotide sensitivity named sgGOLDFISH that consists of enhanced Cas9 (eCas9) nickase, a single 5' extended guide RNA (sgRNA), superhelicase and a set of fluorescently-labeled single-stranded oligonucleotide

probes. To image a specific genomic locus, eCas9 and sgRNA are first applied to create a single-stranded break at the target site. This is followed by loading of superhelicase that locally unwinds the DNA duplex and binding of the resulting single-stranded sites to oligonucleotide probes. Single-nucleotide sensitivity is achieved through comparison of labeling efficiency between sgGOLDFISH systems that incorporate sgRNAs containing one and two mismatches against a given protospacer. In general, I feel the present work serves well as a nice extension of the authors' previous development "GOLDFISH". However, the experimental setup and data interpretation of the present study suffer from several limitations, making the usefulness and generalizability of sgGOLDFISH highly questionable. Additionally, I would add that the manuscript was written in a rather disorganized fashion, and some important key principles behind the experimental setup were not clearly described. My specific concerns are:

1. This work is based upon the fact that eCas9 cannot differentiate perfect targets and targets having a single mismatch well, but can readily differentiate the latter and targets having two mismatches. Thus, the authors used an sgRNA having one mismatch to image the perfect target (e.g., Lamin A) and an sgRNA having two mismatches to image the target with a single mismatch (e.g., progerin). These points need to be clearly stated in the abstract and again somewhere in the manuscript body. Proper references should also be provided.
2. Background information should be provided before discussing the experimental setup and results. For example, the comparison of sgGOLDFISH performance in proteinase-treated versus untreated cells made sense only after reading lines 141-142. In addition, the rationale behind the use of 5'extended guide RNA as opposed to standard guide RNA should be discussed (line 55).
3. It is not clear that why the authors compared sgGOLDFISH genomic labeling between HGPS cells and ABE-corrected HGPS cells rather than between HGPS cells and wildtype cells.
4. The authors should perform dual-color sgGOLDFISH to localize the Lamin A and progerin genes simultaneously rather than single-color sgGOLDFISH for each gene (Fig 2).
5. Inadequate validation methods. Progerin immunofluorescence (IF) is not an appropriate approach to validate whether the bright spots represent progerin loci. A previously-reported FISH approach (e.g., CASFISH or GOLDFISH) should be used for validation. In particular, a target locus correctly labeled by sgGOLDFISH should colocalize better with a FISH-labeled adjacent region than with a FISH-labeled distal region.
6. In line with 5, additional spots/cloud-like patterns in the sgGOLDFISH channel are readily observed in several cells (Figure 1C, Extended Data Fig. 6). Are they nonspecific signals?

Reviewer #3:

Remarks to the Author:

The authors describe a new method, sgGOLDFISH that allows direct visualization of point mutations in situ. This new method is built upon the author's recently published technique, GOLDFISH, which images non-repetitive genomic loci utilizing multiple gRNAs. Overall the technique is clever and could be very useful in some contexts where the SNV is within a certain defined distance from the PAMM. The new method incorporates a few different concepts from additional manuscripts from this research group. However, this is also a weakness in some regards. As written, the format (condensed with most data as extended figures) and descriptions (rather simplified and too much jargon) are confusing and in some respect assumes that the reader is well versed in the authors prior work. I think that making this into a two figure paper with so much extended data is huge mistake. I would recommend significant revisions in the format with much more descriptions of the rationale and experimental details in the main text.

- 1.) Extended Figure 2e is mislabeled, gMUC4-I contains 2 mismatches, while gMUC4-J only contains 1 mismatch compared to extended Figure 2C listing the guide sequences

2.) The assays in the main text and extended data are very poorly described (eg. significant slang and jargon) making it very difficult for the reader to assess the quality and interpretation of the data. The manuscript is written in manner that assumes the readers are experts in the author's previous papers. For example on page on page 3 the authors state "Such specificity was not observed with canonical guide RNA in complex with eCas9 (Extended Data Fig. 2e)". This could have been significantly improved by just adding a bit more information such as ...with a canonical 20-nt protospacer lacking the extra 5' G.....

3.) Extended Figure 3C is very difficult to understand. The authors should figure out a better way to represent the differences of this key piece of data.

4.) Extended Figures 4 & 5 should be referenced in the main text as controls in addition to the supplemental methods describing their new SSB-sensitive ddPCR.

5.) The authors are basing the entire set of assays on their difference of detecting 1 versus two mismatches in the guide but the written rationale for setting up their assays in this manner is non-existent and only very briefly described in the main text as referenced to Figure 1. I eventually understood the authors intent after studying Figure 1 and repeated reading of the rest of the paper. But this should really just be laid out and described in a much more thorough way in the main text when initially describing the technique of using a two mismatched guide to assay for loss of cleavage in the case of a SNV.

6.) The authors should also report the average number of MUC4-R foci/cell

7.) For figure 1B/C, it is not clear what the zoomed in figures are showing. I can assume (Figure 1B) that you are showing no co-localization of MUC4-NR, one allele co-localization, both alleles co-localized. But adding simple labels will help the reader tremendously and remove any ambiguity, particularly for color blind readers such as myself.

8.) Why are there three foci in two of the zoomed panels in Figure 1C?

9.) It would be beneficial to briefly describe the adenine base editor ABE7.10max-VRQR (ABE) mRNA system rather than simply referencing the technology. Perhaps introduce why traditional methods run the risk of having unwarranted DNA insertions and more importantly how this could affect your new method before describing how the ABE system circumvents this problem.

Point-by-point response to the reviewers' comments

Reviewers' Comments to the Authors

Reviewer #1 (Remarks to the Author):

In this manuscript Taekjip Ha and colleagues describe a variant of their technique called GOLDFISH that was described earlier in Wang, Y. et al. *Molecular Cell* (2021). In the present manuscript they apply it for point mutation detection. In this technique a Cas9-nickase is used to bind to the perfect target but not to a mismatched target. The nicked strand is displaced by a helicase and the exposed target strand is then detected using smFISH probes. In order to achieve the required specificity, the guide RNAs are designed so that they contain an additional mismatch against the wild-type (and mutant) sequence as a result, the mutant target is mismatched at one position and the wild-type is mismatched at two. Combined specificity of Cas9 cleavage with smFISH is a great strength. Convincing results are shown for two SNPs. The paper is well-written and the quality of the work appears to be high. Here are some concerns (not in the order of significance):

1. Efficiency of detection is 26% which is disappointing. This may severely impact the utility of the technique, especially when more than one SNP will need to be detected from the same cell. Imagine you need to detect two SNPs in the same cell, if the probability of detection of one is 26%, the probability of detection of both will be only 7%. Also, if you are imaging a tissue section for just one SNP, 75% of the cells will be missed. This is not the case with some of the competing technologies, such as ampFISH (Marras et al *PNAS*, 2019), which even though has similarly low detection efficiency per target mRNA molecule, it will classify the cell correctly because many mRNA molecules are present in each cell

The reviewer made a good point that SNV-sensitive RNA FISH techniques are able to detect SNVs in actively transcribed genes because each cell will have many copies of RNA molecules. We revised the Discussion section (6th paragraph) to make this point clear.

2. Related to this, what do the authors think is the reason for low efficiency detection? Is Cas9 not binding because of the additional mismatch, or is the Cas9 is inherently inefficient? The use of a repeated sequence as a standard to locate all copies the MUC4 locus has worked out well in this system, but that is not a general case.

We added a few sentences to discuss the the reason for low efficiency detection in the 6th paragraph of the Discussion section:

“The detection efficiency of sgGOLDFISH is determined by eCas9 nickase RNP binding and cleavage efficiency, Rep-X unwinding efficiency, and probe hybridization efficiency. In our in vitro cleavage assays (Fig. 1c and Supplementary Fig. 6), adding 50 times excess of the eCas9 RNP in molar ratio cannot completely cleave its DNA substrates, suggesting the intentionally introduced mismatch in guide RNA likely reduces the DNA cleavage efficiency. The local environment such as nucleosomes may also hinder the binding of eCas9 nickase RNP and the DNA unwinding of Rep-X, therefore reduces detection efficiency of sgGOLDFISH. Given that tens of probes tile the target DNA sequence

in sgGOLDFISH, probe hybridization efficiency should not be a major limiting factor in sgGOLDFISH.”

During the development of sgGOLDFISH, we used the MUC4 repetitive locus to confirm that the sgGOLDFISH signals of the MUC4 non-repetitive locus (MUC4-NR) are colocalized with the GOLDFISH signals from the MUC4 repetitive locus, and to estimate the labeling efficiency of MUC4-NR sgGOLDFISH. The sgGOLDFISH method itself does not require a repeated sequence nearby as a standard to locate all copies.

3. In order to evaluate the performance of the second system, LMNA locus, the author should provide its copy number status, does it have two copies or more?

The HEK293FT cells used for this study appear to contain three copies of chromosome 1 (Supplementary Fig. 5a). LMNA is located on the chromosome 1, which could explain that we observed up to 5 LMNA foci per cell (Fig. 3e). We also mentioned this in the Results section in the revised manuscript:

“Note that the LMNA and ACTB genes are on chromosome 1 (three copies, Supplementary Fig. 5a) and chromosome 7 (three copies, Supplementary Fig. 5a), respectively.”

4. A possible weakness is that only one variant is reported by the system. The competing technology, ampFISH reports wild-type in one color and the mutant in the second one. Therefore wild-type, mutant and heterozygotes can be distinguished for any expressing gene locus.

We thank the Reviewer for noting this limitation, and we added a sentence in the 6th paragraph of the Discussion section to make this point clear to readers:

“Because sgGOLDFISH probes are designed to bind the genomic regions downstream of both the wild-type and point-mutated sequences, with the eCas9 nickase cleavage differentiating the wild-type and point-mutated sequences, sgGOLDFISH cannot concurrently label the wild-type allele and mutant allele with probes of different colors.”

5. Another possible concern is how general is this technique given that the Cas9 system is rather tolerant of mismatches, even more than one (<https://www.nature.com/articles/nbt.2675>). How will the design for a new SNP work? Where would one place the deliberate mismatch in relation to the intended SNP, and where would the intended SNP be located? Also, doesn't the necessity of the PAM sequence put a significant constrain on what SNP's are detectable?

Indeed, the wild type Cas9 is rather promiscuous in its cleavage activity, likely because it evolved in bacteria with a genome size 1000 fold smaller than mammalian genomes where there was probably little selective pressure to make it more specific. Therefore, we chose to use the specificity-improved Cas9 RNP variant for sgGOLDFISH applications. We added detailed discussion about the design for a new SNP in the Discussion section:

“A previous study demonstrated SNV-detection using Cas9 when the target SNV is located in the PAM proximal region (< 10 base pairs from PAM)¹¹. To fill the gap of PAM-distal SNV detection,

here we focused on imaging of SNVs in the PAM-distal region (10 to 20 base pairs from PAM) using sgGOLDFISH. In principle, sgGOLDFISH should be able to detect PAM-proximal SNVs as well. Because Cas9 is more sensitive to PAM-proximal mismatches than PAM-distal mismatches^{38, 39}, SNV-sensitive Cas9 RNP variants should also be able to differentiate DNA targets that differ by a PAM-proximal SNV. Future studies are needed to validate this point.

In this study, the guide RNA of sgGOLDFISH carries an intentionally introduced mismatch at the position immediately next to the intended SNV position. Placing the intentionally introduced mismatch at other positions within the PAM-distal region is likely to work as well because eSpCas9(1.1) has minimal gene editing efficiency at doubly mismatched targets even if two mismatches are not adjacent¹⁷.

The sgGOLDFISH takes advantage of the SNV-sensitive cleavage activity of specificity-improved Cas9 RNP variants. The eCas9 RNP was SNV-sensitive in vitro for the four loci tested in this study (Fig. 1c and Supplementary Fig. 6). However, because the cleavage activity of Cas9 is dependent on local sequence⁴⁰, it is possible that the cleavage activity of eCas9 RNP is not optimal for another SNV target, and the design of guide RNA may need to be adjusted. For example, if the protospacer sequence of a new target has higher than 80% GC content, then Cas9 RNP is likely to have low activity on this target⁴⁰ and a single mismatch may completely abolish the cleavage activity of Cas9 RNP on this target. In this case, the intentionally introduced mismatch should be removed from the guide RNA. The activity of Cas9 RNP can be fine-tuned by adding/removing/adjusting the intentionally introduced mismatch in the guide RNA. Adding and removing the 5' extended guanine on the guide RNA can also reduce or increase the cleavage activity²¹. Currently available specificity-improved Cas9 variants other than eSpCas9(1.1) may also work for sgGOLDFISH, adding flexibility for assembling SNV-sensitive Cas9 RNP with optimal activity and specificity against target SNVs^{18, 19, 20}.

Because the necessity of the PAM sequence may put a constraint on SNVs detectable by sgGOLDFISH, we examined 39 prostate cancer-associated single nucleotide polymorphisms (SNPs) that were reported in a recent study⁴¹. Among the 39 SNPs, 27 SNPs are located to a potential PAM sequence so that they can be placed within the PAM-distal region of the corresponding protospacer, and 35 SNPs can be placed with the protospacer. This suggests that PAM sequence does not put a significant constraint on the targeting scope of SNV.”

6. In order to show generality of the approach, I suggest inclusion of several other unique SNP's if they can be included without significant new work.

We have included two additional targets (Fig.3d and Supplementary Fig. 8). Among the four loci tested (*MUC4*, *LMNA*, *ACTB* and *CDKN2A*), three of them (*MUC4*, *LMNA*, *ACTB*) show strong labeling. We saw very low labeling efficiency for *CDKN2A*, and discussed possible reasons in the main text:

“For the CDKN2A sgGOLDFISH (Supplementary Fig. 7c), we observed minimal labeling efficiency even though we used the fully matched guide RNA (Supplementary Fig. 8a and 8b). We hypothesized that the minimal labeling efficiency at this site is because the eCas9 nickase RNP cannot efficiently cleave the CDKN2A protospacer in fixed cells. To test this hypothesis, the SSB-ddPCR assay was performed

using either eCas9 nickase or dCas9 in complex with the gCDKN2A (Supplementary Fig. 8c). We observed no significant differences between the ddPCR results from eCas9 nickase and dCas9 (Supplementary Fig. 8d and 8e), suggesting that the eCas9 nickase RNP did not cleave the CDKN2A protospacer, probably because local environments such as nucleosomes hinder the eCas9 nickase RNP from targeting its protospacer in the fixed cells.”

7. The point about bulky amplification products in other methods is not so strong, given that the length of rolling circle product is controllable by the time of incubation. In fact, for the routine implementation of the sgGOLDFISH in tissue sections, it will be necessary to have brighter foci than produced by 23 probes and Rolling circle amplification will do better. Also, please site the paper where the idea of probe-tiling was first described.

We agree that by shortening incubation time, the rolling circle product could be smaller. We removed the sentence about bulky amplification products in the first paragraph of the revise main text. Designing more probes (e.g., 36 probes were designed for the 1.5kb LMNA target region) and attaching more than one dyes on each probe (although probe synthesis will be more difficult) could further improve the brightness of sgGOLDFISH signals. To our knowledge, the idea of singly labeled short probe-tiling was first described for imaging individual mRNA by Raj et al in 2008, which is cited in the first paragraph of the revise main text.

8. It is great that they included a comparison with other techniques in supplementary table 1. However, it doesn't include more modern techniques like ampFISH, which was designed for RNA but has been implemented for gene locus identification for expressing genes (Fraser et al PNAS 2021). By saying “SNV-sensitive RNA FISH requires the target RNA to be actively transcribing, thereby excluding nongenic regions and inactive or stochastically expressed genes” early in discussion they are bypassing a comparison with RNA detection techniques.

We revised the supplementary table 1 to include amp-FISH. We also revised the Introduction section to mention RNA detection techniques could also be used for gene locus identification:

“One of the RNA FISH methods, amp-FISH, has also been implemented to identify gene loci carrying SNVs by targeting the nascent RNA clusters of expressed genes⁹.”

Reviewer #2 (Remarks to the Author):

The authors described an in situ genomic imaging approach with single-nucleotide sensitivity named sgGOLDFISH that consists of enhanced Cas9 (eCas9) nickase, a single 5' extended guide RNA (sgRNA), superhelicase and a set of fluorescently-labeled single-stranded oligonucleotide probes. To image a specific genomic locus, eCas9 and sgRNA are first applied to create a single-stranded break at the target site. This is followed by loading of superhelicase that locally unwinds the DNA duplex and binding of the resulting single-stranded sites to oligonucleotide probes. Single-nucleotide sensitivity is achieved through comparison of labeling efficiency between sgGOLDFISH systems that incorporate sgRNAs containing one and two mismatches against a given protospacer. In general, I feel the present work serves well as a nice

extension of the authors' previous development "GOLDFISH". However, the experimental setup and data interpretation of the present study suffer from several limitations, making the usefulness and generalizability of sgGOLDFISH highly questionable. Additionally, I would add that the manuscript was written in a rather disorganized fashion, and some important key principles behind the experimental setup were not clearly described.

My specific concerns are:

1. This work is based upon the fact that eCas9 cannot differentiate perfect targets and targets having a single mismatch well, but can readily differentiate the latter and targets having two mismatches. Thus, the authors used an sgRNA having one mismatch to image the perfect target (e.g., Lamin A) and an sgRNA having two mismatches to image the target with a single mismatch (e.g., progerin). These points need to be clearly stated in the abstract and again somewhere in the manuscript body. Proper references should also be provided.

We have revised the abstract and main text to make these points clear. In the abstract, we added:

"The guide RNA carries an intentionally introduced mismatch so that while wild-type target DNA sequence can be efficiently cleaved, a mutant sequence with an additional mismatch (e.g., caused by a point mutation) cannot be cleaved. Because sgGOLDFISH relies on genomic DNA being cleaved by Cas9 to reveal probe binding sites, the probes will only label the wild-type sequence but not the mutant sequence. Therefore, sgGOLDFISH has the sensitivity to differentiate the wild-type and mutant sequences differing by only a single base pair."

In the 2nd paragraph of the Results section, we described these points again:

"Specificity-improved Cas9 variants generally cleave singly mismatched targets as well as their cognate DNA target, but they do not cleave DNA target with two or more mismatches^{17, 18, 19}. We reasoned that if a mismatch is intentionally introduced to the guide RNA, any additional mismatch on the DNA target, such as an SNV, would abolish cleavage activity of the specificity-improved Cas9 RNP variants. Therefore, a specificity-improved Cas9 variant in complex with guide RNA that carries an intentionally introduced mismatch should be SNV-sensitive."

2. Background information should be provided before discussing the experimental setup and results. For example, the comparison of sgGOLDFISH performance in proteinase-treated versus untreated cells made sense only after reading lines 141-142. In addition, the rationale behind the use of 5' extended guide RNA as opposed to standard guide RNA should be discussed (line 55).

We added more background information in the revised manuscript. For the two specific points the reviewer pointed out, we added background information about proteinase treatment in the Introduction section:

"The SNV-sensitive nuclear DNA FISH techniques involve proteinase treatment^{11, 12}, making them incompatible with immunofluorescence imaging of proteins." We also added the rationale behind

the use of 5' extended guide RNA in the 2nd paragraph of the Results section: *"Extending the 5' end of a canonical 20-nt protospacer crRNA with an extra guanine (G) in a guide RNA (hereinafter called 5' extended guide RNA, Supplementary Fig. 2a) also enhances DNA cleavage specificity²¹."* We found even if there are two mismatches, eCas9 in complex with standard guide can still cleave significant fraction of DNA, as we mentioned in the 3rd paragraph of the Results section: *"We observed efficient cleavage for 4 out of 5 guide RNAs with 1 PAM-distal mismatch, but no cleavage activity for five guide RNAs containing 2 PAM-distal mismatches (Fig. 1b and 1c). Such specificity was not observed with a canonical 20-nt protospacer guide RNA in complex with eSpCas9(1.1) (Supplementary Fig. 2c)."*

3. It is not clear that why the authors compared sgGOLDFISH genomic labeling between HGPS cells and ABE-corrected HGPS cells rather than between HGPS cells and wildtype cells.

Using ABE-corrected HGPS cells rather than wild-type cells is more therapeutic-relevant. The HGPS patients have only HGPS cells that carries the pathological point mutation in *LMNA* gene. ABE can correct a fraction of HGPS cells *in vivo*, which can ameliorate some HGPS features in mice (Koblan, Luke W., et al. Nature, 2021). If ABE is used to treat a HGPS patient, the patient will have mixed cell populations of HGPS cells and ABE-corrected HGPS cells.

By identifying HGPS cells or ABE-corrected HGPS cells from a mixed cell population, we also demonstrated sgGOLDFISH can be used to detect base editing outcomes at the single cell resolution.

4. The authors should perform dual-color sgGOLDFISH to localize the Lamin A and progerin genes simultaneously rather than single-color sgGOLDFISH for each gene (Fig 2).

Cas9 and probes are designed to bind identical sites, with the Cas9 cleavage differentiating the wild-type and mutant *LMNA* alleles (Fig. 1a). For this reason, the dual color sgGOLDFISH the reviewer suggested is not possible. This point is mentioned in the 6th paragraph in the Discussion section where we discuss the limitations of our method:

"Because sgGOLDFISH probes are designed to bind the genomic regions downstream of both the wild-type and point-mutated sequences, with the eCas9 nickase cleavage differentiating the wild-type and point-mutated sequences, sgGOLDFISH cannot concurrently label the wild-type allele and mutant allele with different colors of probes."

5. Inadequate validation methods. Progerin immunofluorescence (IF) is not an appropriate approach to validate whether the bright spots represent progerin loci. A previously-reported FISH approach (e.g., CASFISH or GOLDFISH) should be used for validation. In particular, a target locus correctly labeled by sgGOLDFISH should colocalize better with a FISH-labeled adjacent region than with a FISH-labeled distal region.

We have performed a validation experiment similar to the reviewer's suggestion where we showed that MUC4-NR region is correctly labeled by sgGOLDFISH. The MUC4-R and MUC4-NR regions are within the MUC4 gene, spaced by ~19kb (Fig. 3a). We show that the signals of MUC4-NR labeled by

sgGOLDFISH are colocalized with the signals of MUC4-R labeled by GOLDFISH (Fig. 3b). In contrast, the MUC4-R GOLDFISH signals and LMNA sgGOLDFISH signals (*LMNA* gene is on chromosome 1) are not colocalized (Supplementary Fig. 10b and 10c).

The progerin locus is the *LMNA* gene with a C-to-T point mutation at c. 1824 position. The progerin locus and Lamin A locus are at the same position (i.e., *LMNA* gene) on chromosome 1. A control FISH-labeled region that colocalizes with the progerin locus will also colocalize with the Lamin A locus. Therefore, the control FISH-labeled region cannot validate that the sgGOLDFISH-based call of the progerin locus or Lamin A wild type locus.

In this study, we utilized two approaches to differentiate HGPS cells and ABE-corrected HGPS cells from the 1:1 mixed cell population: sgGOLDFISH and progerin immunofluorescence (IF) (Fig. 4d and 4e and Supplementary Fig. 10a and 10b). When using sgGOLDFISH, a cell with at least one progerin sgGOLDFISH spot should be a HGPS cell. A cell with two or four (depending on cell cycle state) Lamin A sgGOLDFISH spots should be an ABE-corrected HGPS cell. When using progerin IF, HGPS cells have higher progerin IF signals than ABE-corrected HGPS cells (see Fig. 4f “Untreated” and Fig. 4g “ABE treated”).

We found that sgGOLDFISH identified HGPS cells (Fig. 4d and 4f “Mutant-positive”) have much higher progerin IF signals compared to sgGOLDFISH identified ABE-corrected cells (Fig. 4e and 4g “Correction-positive”), which suggests that sgGOLDFISH correctly differentiated HGPS cells and ABE-corrected HGPS cells from the 1:1 mixed cell population.

6. In line with 5, additional spots/cloud-like patterns in the sgGOLDFISH channel are readily observed in several cells (Figure 1C, Supplementary Fig. 6). Are they nonspecific signals?

Cloud-like spots are nonspecific background signals. These are distinguishable in intensity and shape from specific sgGOLDFISH spots.

Reviewer #3 (Remarks to the Author):

The authors describe a new method, sgGOLDFISH that allows direct visualization of point mutations in situ. This new method is built upon the author’s recently published technique, GOLDFISH, which images non-repetitive genomic loci utilizing multiple gRNAs. Overall the technique is clever and could be very useful in some contexts where the SNV is within a certain defined distance from the PAM. The new method incorporates a few different concepts from additional manuscripts from this research group. However, this is also a weakness in some regards. As written, the format (condensed with most data as extended figures) and descriptions (rather simplified and too much jargon) are confusing and in some respect assumes that the reader is well versed in the authors prior work. I think that making this into a two figure paper with so much extended data is huge mistake. I would recommend significant revisions in the format with much more descriptions of the rationale and experimental details in the main text.

1.) Extended Figure 2e is mislabeled, gMUC4-I contains 2 mismatches, while gMUC4-J only contains 1 mismatch compared to extended Figure 2C listing the guide sequences

This mislabeling has been corrected in the revised manuscript.

2.) The assays in the main text and extended data are very poorly described (eg. significant slang and jargon) making it very difficult for the reader to assess the quality and interpretation of the data. The manuscript is written in manner that assumes the readers are experts in the author's previous papers. For example on page on page 3 the authors state "Such specificity was not observed with canonical guide RNA in complex with eCas9 (Supplementary Fig. 2e)". This could have been significantly improved by just adding a bit more information such as ...with a canonical 20-nt protospacer lacking the extra 5' G.....

We added more background information and description of experimental setup in the revised manuscript, including the specific example that the reviewer pointed out.

3.) Extended Figure 3C is very difficult to understand. The authors should figure out a better way to represent the differences of this key piece of data.

We have made substantial revision on the text describing that piece of data to make it easier to understand (the "SSB-ddPCR for nicking efficiency quantification" subsection in the Results section).

4.) Extended Figures 4 & 5 should be referenced in the main text as controls in addition to the supplemental methods describing their new SSB-sensitive ddPCR.

We revised the manuscript as the reviewer suggested.

5.) The authors are basing the entire set of assays on their difference of detecting 1 versus two mismatches in the guide but the written rationale for setting up their assays in this manner is non-existent and only very briefly described in the main text as referenced to Figure 1. I eventually understood the authors intent after studying Figure 1 and repeated reading of the rest of the paper. But this should really just be laid out and described in a much more thorough way in the main text when initially describing the technique of using a two mismatched guide to assay for loss of cleavage in the case of a SNV.

We have revised the abstract and main text to make these points clear. In the abstract, we added:

"The guide RNA carries an intentionally introduced mismatch so that while wild-type target DNA sequence can be efficiently cleaved, a mutant sequence with an additional mismatch (e.g., caused by a point mutation) cannot be cleaved. Because sgGOLDFISH relies on genomic DNA being cleaved by Cas9 to reveal probe binding sites, the probes will only label the wild-type sequence but not the mutant sequence. Therefore, sgGOLDFISH has the sensitivity to differentiate the wild-type and mutant sequences differing by only a single base pair."

In the 2nd paragraph of the Results section, we described these points again:

“Specificity-improved Cas9 variants generally cleave singly mismatched targets as well as their cognate DNA target, but they do not cleave DNA target with two or more mismatches^{17, 18, 19}. We reasoned that if a mismatch is intentionally introduced to the guide RNA, any additional mismatch on the DNA target, such as an SNV, would abolish cleavage activity of the specificity-improved Cas9 RNP variants. Therefore, a specificity-improved Cas9 variant in complex with guide RNA that carries an intentionally introduced mismatch should be SNV-sensitive.”

6.) The authors should also report the average number of MUC4-R foci/cell

We report the average number and histogram of MUC4-R foci/cell in Supplementary Fig. 5b.

7.) For figure 1B/C, it is not clear what the zoomed in figures are showing. I can assume (Figure 1B) that you are showing no co-localization of MUC4-NR, one allele co-localization, both alleles co-localized. But adding simple labels will help the reader tremendously and remove any ambiguity, particularly for color blind readers such as myself.

We added labels on the figure and additional description in the Fig. 3b legend:

“Single cells outlined in orange are magnified on the upper-right corner, which show (i.) two detected MUC4-NR alleles (ii.) one detected MUC4-NR allele (iii.) no detected MUC4-NR allele.”

8.) Why are there three foci in two of the zoomed panels in Figure 1C?

There are three copies of chromosome 1 (*LMNA* gene is on chromosome 1) in the HEK293FT cells (Supplementary Fig. 5a). So, a cell could have up to 6 copies of *LMNA* foci in the S/G2/M phase.

9.) It would be beneficial to briefly describe the adenine base editor ABE7.10max-VRQR (ABE) mRNA system rather than simply referencing the technology. Perhaps introduce why traditional methods run the risk of having unwarranted DNA insertions and more importantly how this could affect your new method before describing how the ABE system circumvents this problem.

We added more description about the ABE mRNA system in the “sgGOLDFISH for Hutchinson-Gilford progeria syndrome mutation” subsection in the Results section:

*“ABE is precise, efficient, and convenient in correcting point mutations compared to traditional methods such as helper-dependent adenoviral vectors and Cas9-mediated gene editing^{31, 32}. Previous studies utilized lentivirus or adeno-associated virus (AAV) to deliver ABE into cells for correcting the HGPS mutation^{30, 33}. The genome integration sites of lentivirus are generally unpredictable³⁴. There are also safety concerns of developing hepatocellular carcinoma in mice after AAV treatment, which is likely due to integration of the AAV genome^{30, 35}. In contrast, non-viral delivery of the ABE in the form of mRNA does not risk unwanted DNA integration into the genome. Transient expression of the ABE due to the finite lifetime of the delivered mRNA could also reduce potential off-target base editing. Constant expression of ABE may also block the point mutation site as ABE can keep binding at the site even after editing, potentially hindering the binding of our eCas9 nickase RNP to the *LMNA* point mutation site during sgGOLDFISH.”*

Reviewers' Comments:

Reviewer #1:

Remarks to the Author:

The authors have responded satisfactorily to my comments and the paper now looks in great shape.

Reviewer #2:

Remarks to the Author:

The authors have satisfactorily addressed my concerns; I therefore recommend this work for publication.

Reviewer #3:

Remarks to the Author:

The authors describe a new method, sgGOLDFISH that allows direct visualization of point mutations in situ. This new method is built upon the author's recently published technique, GOLDFISH, which images non-repetitive genomic loci utilizing multiple gRNAs. The new method incorporates a few different concepts from additional manuscripts from this research group. Overall the technique is clever and could be very useful in some contexts where the SNV is within a certain defined distance from the PAMM. This method differs from SNV-sensitive RNA FISH since it allows the researchers to examine SNVs in intergenic and lowly expressed genes. sgGOLDFISH is also compatible with immunofluorescence, which is not possible with SNV-sensitive DNA FISH. The methodology is sound with conclusions that are well supported by the data. The revised manuscript has now corrected many of the flaws of the prior submission and provides enough detail for the work to be reproduced.

Point-by-point response to the reviewers' comments

REVIEWERS' COMMENTS

Reviewer #1 (Remarks to the Author):

The authors have responded satisfactorily to my comments and the paper now looks in great shape.

Reviewer #2 (Remarks to the Author):

The authors have satisfactorily addressed my concerns; I therefore recommend this work for publication.

Reviewer #3 (Remarks to the Author):

The authors describe a new method, sgGOLDFISH that allows direct visualization of point mutations in situ. This new method is built upon the author's recently published technique, GOLDFISH, which images non-repetitive genomic loci utilizing multiple gRNAs. The new method incorporates a few different concepts from additional manuscripts from this research group. Overall the technique is clever and could be very useful in some contexts where the SNV is within a certain defined distance from the PAMM. This method differs from SNV-sensitive RNA FISH since it allows the researchers to examine SNVs in intergenic and lowly expressed genes. sgGOLDFISH is also compatible with immunofluorescence, which is not possible with SNV-sensitive DNA FISH. The methodology is sound with conclusions that are well supported by the data. The revised manuscript has now corrected many of the flaws of the prior submission and provides enough detail for the work to be reproduced.

RESPONSE TO THE REVIEWERS' COMMENTS

We appreciate the time and effort of the reviewers in providing valuable feedback on our manuscript.